

# A novel user-centric happiness model for personalized tour recommendations

Mohammed Alatiyyah

Department of Computer Science, College of Computer Engineering and Sciences, Prince Sattam Bin Abdulaziz University, Al-kharj, Saudi Arabia

## ABSTRACT

A novel personalized tour recommendation model, the Happiness Model (HM), is presented. The HM optimizes itineraries by considering traveler satisfaction as a function of time and maximizing it over the trip duration. The model integrates the Item Constraints Data Model (ICDM) to reduce data dimensionality and search space. By considering various activities within different points of interest (POIs) and minimizing wasted time, the HM overcomes the limitations of existing methods. Unlike existing POI-centric models, the HM is time-centric, creating tour recommendations that maximize user satisfaction throughout the trip. Experimental results demonstrate the model's effectiveness in generating personalized tour recommendations aligned with user preferences. The HM achieves an average satisfaction score of 0.85 across multiple datasets, outperforming traditional models such as the Time-Dependent Orienteering Problem with Time Windows (TOPTW), which achieves an average score of 0.72. Additionally, the HM reduces waiting times by 30% and increases the number of recommended POIs by 20% compared to existing methods. These results highlight the HM's ability to provide more efficient and enjoyable travel experiences.

# INTRODUCTION

## Research background

Recently, governments, industry stakeholders, and academic researchers in tourism studies have progressively focused on happiness research to address their respective challenges (*Chen & Li, 2018*). Travelers often embark on vacations with the aim of enhancing their happiness levels, and recent findings highlight the positive impact of holidays on individual well-being (*Nawijn, 2011*). However, existing travel recommender systems (TRSs) often fail to fully capture the complexities of traveler preferences, particularly in terms of time management, activity selection, and satisfaction optimization.

## Research problem

Travel-related decision-making involves consideration of multiple objectives and constraints. According to *Burke & Kendall (2014)*, higher-quality personalized recommendations significantly persuade travelers to undertake specific trips. Essentially, aligning suggested tours with traveler preferences is found to be a key factor in fostering

Corresponding author
Mohammed Alatiyyah,
M.alatiyyah@psau.edu.sa

trust towards recommender systems. Current TRSs predominantly focus on selecting points of interest (POIs) based on static preferences, neglecting critical factors such as time constraints, waiting times, and the holistic satisfaction of travelers throughout their trips. This results in suboptimal itineraries that do not align with the dynamic needs of modern travelers. Additionally, existing models often treat waiting times and travel connections as fixed costs, failing to account for their impact on overall traveler satisfaction (*Halder et al., 2024*). These limitations highlight the need for a more comprehensive and time-centric approach to tour recommendations.

To further elucidate the research problem addressed in this article, an illustrative example is presented involving two distinct groups of travelers who are visiting a city, denoted as A and B. Figure 1 illustrates a city map with various POIs and designated starting and ending points. The numbers on the map represent unique identifiers for each POI. The starting point is labeled as 'Start Point (Hotel)', and the ending point is labeled as 'End Point (Airport)'. POIs are categorized based on their appeal to two distinct traveler groups: Group A (preferring natural environments such as zoos, parks, and mountains) and Group B (preferring modern settings such as shopping malls, street markets, and museums). The edges between POIs represent travel routes, with associated costs or distances indicated by the numbers. This figure serves as a visual representation of the input data used to generate personalized tour recommendations. Moreover, it is not merely the type of POIs that differentiates the groups; their preferred modes of transportation between POIs also vary significantly. For instance, Group B favors walking between POIs to experience more traditional areas and street markets. In contrast, Group A prefers traveling by car, which enables them to visit a greater number of natural POIs, such as wildlife parks and rivers. This variation in travel preferences underscores the need for customized trip planning to accommodate the distinct requirements of each group.

## Proposed solution

To address these challenges, this article introduces the Happiness Model (HM), a novel personalized tour recommendation model that optimizes itineraries by maximizing traveler satisfaction as a function of time. The HM integrates the Item Constraints Data Model (ICDM) to reduce data dimensionality and search space, enabling efficient and personalized recommendations. Unlike traditional POI-centric models, the HM adopts a time-centric approach, dynamically adjusting recommendations to maximize satisfaction over the entire trip duration. The model considers three key components: (1) activities at POIs, (2) connections between POIs, and (3) waiting times, ensuring a holistic and efficient travel plan.

## Context and motivation

Given the growing importance of time management in travelers' hectic schedules— wherein they frequently coordinate bookings for hotels, flights, tours, and event tickets simultaneously—there is a pressing need for a model capable of integrating these factors to provide comprehensive trip plan recommendations. Recommender systems (RSs) have

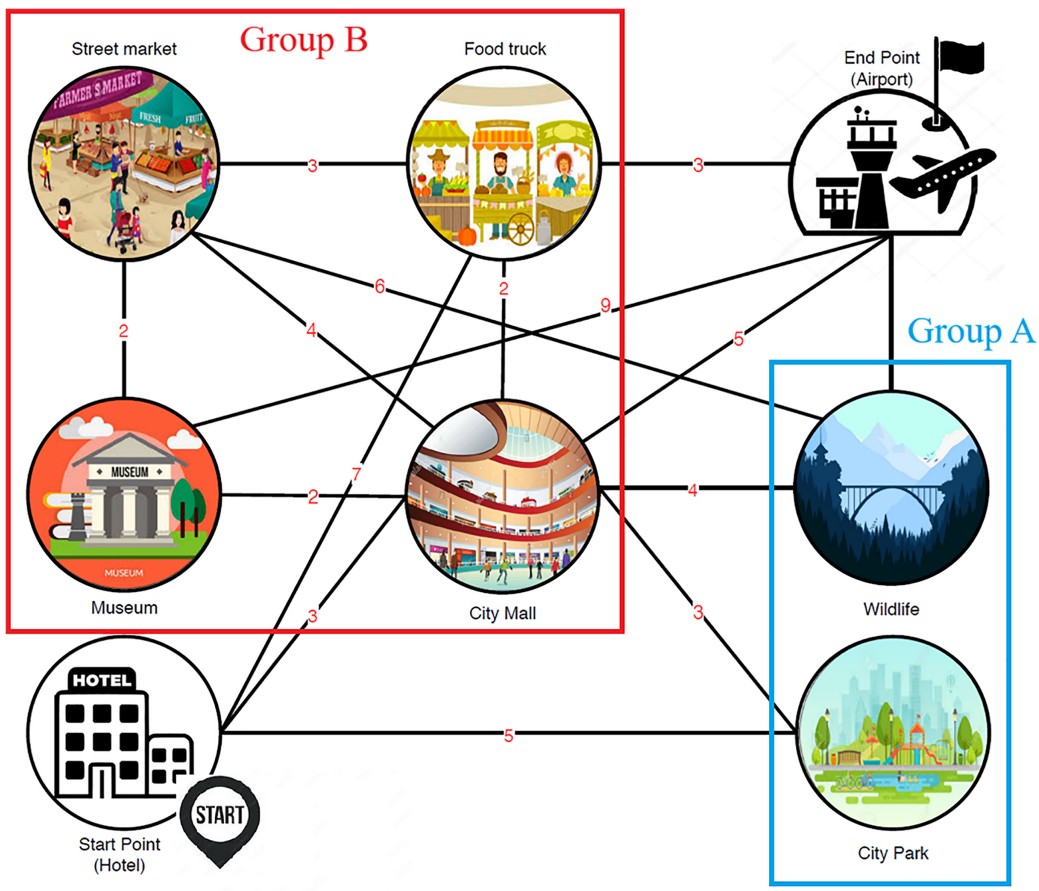

**Figure 1 City map with POIs and designated start/end points.**

emerged as a crucial tool in aiding users by providing personalized suggestions amidst extensive selections of options. This capability is anchored in decision-support mechanisms (*Mao et al., 2024*; *Wu, Lyu & Liu, 2022*; *Felfernig et al., 2018*; *Liu, Zhong & Zhou, 2022*). Specifically, travel recommender systems (TRSs) endeavor to facilitate itinerary planning by matching user preferences with the attributes of their intended journeys (*Zhou et al., 2023*). Nonetheless, empirical evidence indicates a bias in current TRSs towards recommending points of interest (POIs) predominantly situated in urban locales (*Quijano-Sánchez et al., 2020*). Effective travel planning demands the consideration of various factors, including temporal constraints, budgetary restrictions, transportation alternatives, and meteorological conditions (*Sun & Wandelt, 2021*).

## Challenges in existing systems

Driven by the economic importance of the tourism industry, computer scientists have been at the forefront of developing innovative tools to enhance the sector. Such tools provide personalized recommendations that cater to users' unique constraints and preferences, potentially enhancing tourist satisfaction and industry growth. This research area has

garnered significant attention from various disciplines, including computer science, information systems, and marketing (*Sarkar et al., 2023*). Travel recommender systems (TRSs) have become a cornerstone of the tourism industry, offering personalized itinerary suggestions (*Sarkar et al., 2023*). These systems leverage various models that consider user preferences and constraints to design optimal tour plans (*Beed et al., 2020*). While relevant studies from literature acknowledge the limitations of solely focusing on POIs, they still face challenges in comprehensively addressing both travel preferences and time management (*Beed et al., 2020*). *Subramaniyaswamy et al. (2018)* propose a multi-modal itinerary recommendation system that considers user preferences for transportation and activity duration. However, their model treats waiting times as a fixed cost between POIs, neglecting the impact of unplanned delays or activity overruns. *Renjith, Sreekumar & Jathavedan (2020)* introduce a context-aware recommendation system that incorporates real-time information like weather and traffic conditions. This approach offers dynamic recommendations but may struggle with limited user data on travel preferences for connection types and their impact on enjoyment. On the other hand, some studies delve deeper into specific aspects beyond POI selection, but lack a holistic approach. *Chen et al. (2017)* present a framework for optimizing travel routes considering user preferences for scenic beauty along the way. While this caters to a specific travel desire, it does not integrate preferences for transportation type or waiting times. *Su et al. (2020)* propose an emotion-based model for personalized travel recommendation systems. This addresses the emotional aspect of travel but does not explicitly consider the impact of connection preferences and waiting times on user emotions.

## Key contributions

In light of the preceding discussion, this article presents a novel personalized tour recommendation model, the HM, that offers the following key contributions:

- Time-centric approach: Unlike traditional models that focus on points of interest (POIs), the proposed HM focuses on maximizing user satisfaction over time, dynamically adjusting to traveler preferences during the trip.
- Integration of the Item Constraints Data Model (ICDM): By incorporating ICDM, the HM reduces data dimensionality, thereby improving computational efficiency while maintaining personalization.
- Consideration of various activities: The proposed HM incorporates activities such as visiting POIs, traveling between POIs, and minimizing wasted time to create more comprehensive itineraries while ensuring a holistic and efficient travel plan.
- Overcoming limitations of existing methods: The proposed HM addresses the shortcomings of current models by considering a wider range of factors that influence traveler satisfaction, such as time management, connection preferences, and waiting times.

The article is organized as follows: "Related Works" provides a review of related work. "Item Constraints Data Model" introduces the Item Constraints Data Model (ICDM).

"The Proposed Happiness Model" presents the proposed Happiness Model (HM), including its mathematical formulation and integration with the ICDM. "Experiments" and "Discussion" details the experimental setup and results. Finally, "Conclusion" concludes the article, summarizing the key findings and discussing future research directions.

**Remark 1.** *The proposed happiness model introduces an innovative approach to assessing travelers' satisfaction levels. It addresses the limitations of existing models, which fail to account for the value of connections and waiting times. Specifically, the HM optimizes tour itineraries based on three key categories of actions: activities, connections, and waiting times. Unlike traditional models that rely on aggregating scores from various points of interest (POIs), the HM tackles the NP-hard orienteering problem (OP) by optimizing a diverse range of trip activities. This approach adds complexity to the model but significantly enhances its ability to generate personalized and efficient itineraries.*

## RELATED WORKS

In the realm of enhancing the traveler's experience, the formulation and optimization of appropriate objective functions are essential for addressing travel recommendation challenges (*Sarkar et al., 2023*). In the domain of TRSs, and particularly within the tour trip design problem (TTDP), a variety of methods and approaches have been advanced (*e.g.*, *Zaizi, Qassimi & Rakrak, 2023*; *Wan et al., 2018*). Nonetheless, a notable limitation of prior research is the emphasis on the selection of POIs without accounting for other significant factors that affect traveler satisfaction (*Asaithambi, Venkatraman & Venkatraman, 2023*). For example, existing models frequently overlook traveler preferences concerning travel time between POIs and the effective utilization of the allocated trip duration. These models predominantly construct routes based solely on the aggregated scores of visited locations, thereby presenting considerable limitations in accurately capturing overall traveler satisfaction (*Chen et al., 2022*).

Recommender systems (RSs) predominantly utilize data mining methodologies; however, they frequently encounter difficulties in generating highly personalized itineraries compared to those developed exclusively through optimization strategies (*Beraldi et al., 2021*). Nevertheless, current models addressing the Tourist Trip Design Problem (TTDP) exhibit deficiencies in meeting traveler requirements such as waiting times, budgetary constraints, and the simplicity of the route (*Yoon & Choi, 2023*). A notable variant of the TTDP is the orienteering problem (OP), which is categorized as NP-hard. Furthermore, alterations to specific constraints within a model can profoundly influence the efficacy of its algorithms, occasionally resulting in their incapacity to resolve the OP (*Mei, Salim & Li, 2016*).

The primary limitation of the multi-objective orienteering problem (MOOP) is its inability to support personalization, as it fails to incorporate waiting time into its model (*Dutta et al., 2020*). In contrast, the proposed HM is specifically designed to integrate time as a critical component in the construction of a tour itinerary. Furthermore, the HM accounts for a broader array of decision-making parameters, including both multi-value

Points of Interest (POIs) and connections, unlike the MOOP, which only considers multi-value POIs. The HM is thus introduced to address the comprehensive set of decision-making criteria for tourists, categorizing these parameters into three main groups: (1) activities, (2) connections, and (3) waiting time. By encompassing all three of these essential constraints, the HM extends the capabilities of existing models.

## Comparison with previous work

To the best of our knowledge, all preceding models concentrate exclusively on the cumulative scores of visited POIs, which constitutes a significant limitation. Table 1 provides a detailed comparison between the proposed Happiness Model (HM) and existing approaches, highlighting the unique advantages of the HM. Firstly, the Happiness Model (HM) and the Multi-Objective Orienteering Problem (MOOP) assign multiple values (MVs) to each node. The HM is specifically developed to attribute multiple values to various aspects of a trip, including points of interest, travel options, and time constraints. Moreover, the Multi-Constraint Team Orienteering Problem with Multiple Time Windows (MCTOPMTW) incorporates multi-attributes to enforce multi-constraints (MC), ensuring that the trip remains within the bounds of these limitations. Furthermore, the MOOP deals with distinct categories, each offering unique benefits, with the goal of maximizing overall benefits across these categories. Additionally, the MCTOPMTW accommodates tags, allowing each point of interest to have multiple attributes.

In contrast, the proposed HM has the ability to manage both categories and attributes by leveraging the strengths of both types of models. Such innovative approach introduces the consideration of multi-values (MVs) between each node, while other models traditionally focus solely on distance. Specifically, the transition between points of interest involves various values, such as time, journey length, and cost. Additionally, the HM combines these values into a single metric that represents the traveler's satisfaction level. Moreover, travelers with reservations (e.g., flights, hotels, or trains) typically prefer using the entire allocated time for a given trip. Therefore, the proposed HM is designed to consider time inefficiency as a crucial factor that likely impacts the traveler's satisfaction level.

The comparative analysis is further expanded to include additional models, such as the Genetic Algorithm-based Tour Recommendation System (GATRS) and the particle swarm optimization-based approach (PSO). Table 1 demonstrates the superiority of the HM in terms of personalization and the consideration of multiple factors such as time and waiting times, which are not addressed by other models like GATRS and PSO.

In light of the preceding discussion, it is evident that the proposed HM addresses the deficiencies inherent in extant models documented in the literature. In alignment with prior research, the proposed model is structured to assess each discrete moment of a trip by taking into account various trip-related actions. Moreover, the HM employs a temporal framework, utilizing an algorithm to optimize activities at each individual moment rather than relying on a cumulative aggregation of events.

**Table 1  Comparison of HM with existing models.**

| Feature | Happiness model (HM) | MOOP | MCTOPMTW | GATRS | PSO |
|---|---|---|---|---|---|
| Time-centric personalization | Yes | No | Partial | No | No |
| Multi-value connection optimization | Yes | No | No | Partial | Partial |
| Waiting time consideration | Yes | No | Partial | No | No |
| Real-time data integration | Planned | No | No | No | No |

## Advantages of the proposed solutions

The HM offers several advantages over existing models:

- Time-centric optimization: By focusing on maximizing satisfaction over time, the HM provides more dynamic and personalized itineraries.
- Comprehensive consideration of factors: The model accounts for activities, connections, and waiting times, addressing limitations of traditional POI-centric approaches.
- Improved efficiency: The integration of the ICDM reduces computational complexity, enabling faster and more scalable recommendations.
- Enhanced personalization: Experimental results demonstrate that the HM outperforms existing models in terms of satisfaction scores, waiting time reduction, and the number of recommended POIs.

## ITEM CONSTRAINTS DATA MODEL

The item constraints data model (ICDM) has been specifically developed to address the shortcomings identified in prior academic studies. This model is designed with precision to manage data alongside constraints, catering to the needs and preferences of users for their travel itineraries. The data is classified into two categories: (1) dynamic data and (2) static data, facilitating the alignment of item information with user-imposed constraints. Static data is defined by its unchanging nature over time, whereas dynamic data consists of variables that fluctuate over time. User-imposed constraints are further categorized into hard constraints (HCs) and soft constraints (SCs). The formal representation of ICDM is as follows: Let $u$ represent a user with $n$ constraints, including both $HC$ and $SC$. $HC$ comprises a set of hard constraints, each denoted as $hc^m \in HC$ where $m = 1, 2, \ldots, |HC|$, and $SC$ encompasses a set of soft constraints, each denoted as $sc^v \in SC$ where $v = 1, 2, \ldots, |SC|$. These constraints are mathematically described by the following equations:

$$HC_{pti} = \prod_{m=1}^{|HC|} hc_{pti}^m \tag{1}$$

$$SC_{pti} = \text{Aggregation methods} \tag{2}$$

$$\sum_{v=1}^{|SC|} W_v = 1 \tag{3}$$

Equation (1) outlines the computation of hard constraints (HCs), represented as $HC_{pti}$, for a specific item $i$ on day $p$ at time $t$. It is crucial that all HCs are simultaneously satisfied; the failure to meet even one HC results in the nullification of the aggregate value. In contrast, soft constraints (SCs) exhibit varying levels of fulfillment, with compliance being optional. Equation (2) explicates the determination of the user's satisfaction degree based on their SCs, denoted as $SC_{pti}$, for item $i$ on day $p$ at time $t$. The closer the result of Eq. (2) is to unity, the higher the satisfaction level, indicating the fulfillment of a greater number of constraints. In Eq. (3), $W_v$ represents the weight of SC #$v$, with the total sum equaling unity. Subsequent formulations integrate these distinct equations into a comprehensive evaluative metric.

$$S_{pti} = HC_{pti} \times SC_{pti}. \tag{4}$$

Equation (4) quantifies the satisfaction level of a user $u$ concerning an item $i$ on day $p$ at temporal coordinate $t$, under the constraints specified by the user.

The core innovation of the proposed model is the integration of various user-specific constraints into a single metric, denoted as $S_{pti}$, which is accomplished by combining Eqs. (1) and (2). This method ensures that all user-imposed constraints are incorporated within $S_{pti}$, thereby facilitating data dimensionality reduction by merging the distinct values of individual constraints into a cohesive representation. Furthermore, the implementation of the ICDM model is expected to enhance the efficiency of the search process due to the reduced necessity for constraint matching with the trip parameters.

## THE PROPOSED HAPPINESS MODEL

The development of the HM has been undertaken to address traveler preferences with the objective of optimizing user satisfaction (*Alatiyyah, 2019*). Initially, the mathematical formulation of the model is delineated, followed by an integration of the HM with the ICDM framework. The discussion section subsequently encapsulates the significance of the HM in relation to its contemporary applications within the tourism sector.

### Mathematical model

The proposed happiness model is formulated as follows:

- *Directed weighted graph:* A directed weighted graph $G = (V, E)$ represents a city with nodes $i \in I$; $i = 1, \ldots, |I|$ (Points of Interest) and attributes $k \in K$; $k = 1, \ldots, |K|$. The edges in $E$ connect these nodes and have values in $r \in R$; $r = 1, \ldots, |R|$.
- *Travel time and time spent:* The travel time between nodes $i, j \in I$ is denoted by $TT_{ij}$, and the time spent at node $i$ denoted by $ST_i$.
- *Starting and ending points:* Let $s$ and $t$ be a starting and terminal nodes, respectively, where $s = 1$ and $t = |I|$, in which the trip duration can span multiple days. Hence, $p \in P$; $p = \{1, \ldots, |P|\}$ denotes the trip days. Furthermore, each trip encompasses a

start and end time within the day $p$, where $t \in p; t = 1, \ldots, |p|$ represents the set of moments in day $p$.

- *Time constraints:* The daily time constraint is represented by $T_{max}$, and $S_{max}$ denotes the maximum value for all actions, scaled from 0 to 1.
- *Decision variables:* $X_{pti}$ represents whether the user remains at node $i$ on day $p$ at time $t$, $Y_{ptij}$ represents whether the user moves from node $i$ to node $j$ on day $p$ at time $t$, and $Z_{pt}$ represents whether the user is waiting on day $p$ at time $t$.
- *Score values:* $S_{pti}$ denotes the score value of activities at node $i$ on day $p$ at time $t$. The value of a connection from node $i$ to node $j$ based on attribute $r$ on day $p$ at time $t$ is denoted by $C_{ptijr}$, and the value of the waiting time on day $p$ at time $t$ is denoted by $W_{pt}$.

## Objective function

The proposed happiness model is formulated as a multi-objective optimization problem aimed at maximizing traveler satisfaction over time. The objective function of the model is to maximize the total score derived from three actions: activity, connection, and waiting. This is represented by Eq. (5), which includes three functions: $f_1(a)$, $f_2(c)$, and $f_3(w)$. The HM is quantified on a scale from 0 to 1, with 1 indicating optimal user satisfaction during the tour.

$$Max\left(\frac{f_1(a) + f_2(c) + f_3(w)}{T_{max} \times S_{max} \times |P|}\right) \tag{5}$$

where $f_1(a)$, $f_2(c)$, and $f_3(w)$ are defined in Eqs. (6)–(8), respectively. The function $f_1(a)$ represents the level of happiness associated with a given activity. It is normalized to a range between 0 and 1 by dividing the raw value by $(T_{max} \times S_{max} \times |P|)$.

$$f_1(a) = \sum_{p=1}^{|P|} \sum_{t=1}^{|p|} \sum_{i=1}^{|I|} X_{pti} \times S_{pti} \tag{6}$$

$f_2(c)$ denotes the satisfaction metric for the connectivity activity. This metric ranges from 0 to 1, achieved by normalizing the value through division by $(T_{max} \times S_{max} \times |P|)$.

$$f_2(c) = \sum_{p=1}^{|P|} \sum_{t=1}^{|p|} \sum_{i=1}^{|I|} \sum_{j=1}^{|I|} \left(Y_{ptij} \times \sum_{r=1}^{|R|} C_{ptijr}\right). \tag{7}$$

The function $f_3(w)$ denotes the level of satisfaction associated with waiting time, which is computed by normalizing the value through division by $(T_{max} \times S_{max} \times |P|)$.

$$f_3(w) = \sum_{p=1}^{|P|} \sum_{t=1}^{|p|} \left(Z_{pt} \times W_{pt}\right). \tag{8}$$

## Constraint functions

The following constraints ensure the feasibility of the solution. Equation (9) imposes a restriction that permits only a single activity at any given moment. Furthermore, Eq. (10) mandates that the journey on each day $p$ commences from $s$, designated as the starting point. Additionally, Eq. (11) guarantees that on each day $p$, the journey concludes at $e$, which represents the terminal point.

$$\sum_{i=1}^{|I|} X_{pti} + \left( \sum_{i=1}^{|I|} \sum_{j=1}^{|I|} Y_{ptij} \right) + Z_{pt} = 1; \forall t = 1, \ldots, |p|; \forall p = 1, \ldots, |P| \tag{9}$$

$$\sum_{j=1}^{|I|} Y_{p11j} = 1; \forall p = 1, \ldots, |P| \tag{10}$$

$$\sum_{i=1}^{|I|-1} \left( \frac{\sum_{t=1}^{|p|} Y_{pti|I|}}{TT_{i|I|}} \right) = 1; \forall p = 1, \ldots, |P|. \tag{11}$$

Equation (12) serves as a constraint to ensure the connectivity of the tour trip, guaranteeing that the travel time and visiting times correspond accurately to the given data, denoted by $TT_{i,j}$ and $ST_i$, respectively. Furthermore, Eq. (12) aligns with the corresponding equations in the OP model.

$$\frac{\sum_{t=n}^{t_1=n+TT_{s,r}} Y_{p,s,r}^t \times \sum_{t=t_1+1}^{t_2=t_1+ST_r} X_{p,r}^t}{TT_{s,r} + ST_r} = \frac{\sum_{t=t_2+1}^{t_3=t_2+TT_{r,m}} Y_{p,r,m}^t}{TT_{r,m}} \leq 1 \tag{12}$$

$$\forall n \in \{1, \ldots, |p-3|\}; \forall p = 1, \ldots, |P|; m \in I; \forall s, m = 1, \ldots, |I|; \forall r = 2, \ldots, |I-1|.$$

## Algorithm

The proposed HM can be solved using a variety of optimization algorithms, such as genetic algorithms, ant colony optimization, or mixed-integer programming. In this article, an ant colony optimization (ACO) algorithm is proposed to solve the HM.

## An ant colony optimization

The ACO algorithm is employed to solve an NP-hard problem. The algorithm iteratively constructs solutions by simulating the behavior of ants searching for optimal paths. At each iteration, ants probabilistically select the next node (POI) based on pheromone levels and heuristic information.

In ACO algorithm, two sequential pheromone update steps are employed to enhance its efficacy. The initial update, termed as the local pheromone update, occurs during the second step (refer to Fig. 2); after an ant is deployed, it evaluates whether it can identify an improved score for the discovered path (refer to Eq. (15)). The subsequent update transpires after all ants have been deployed (refer to Eq. (17)).

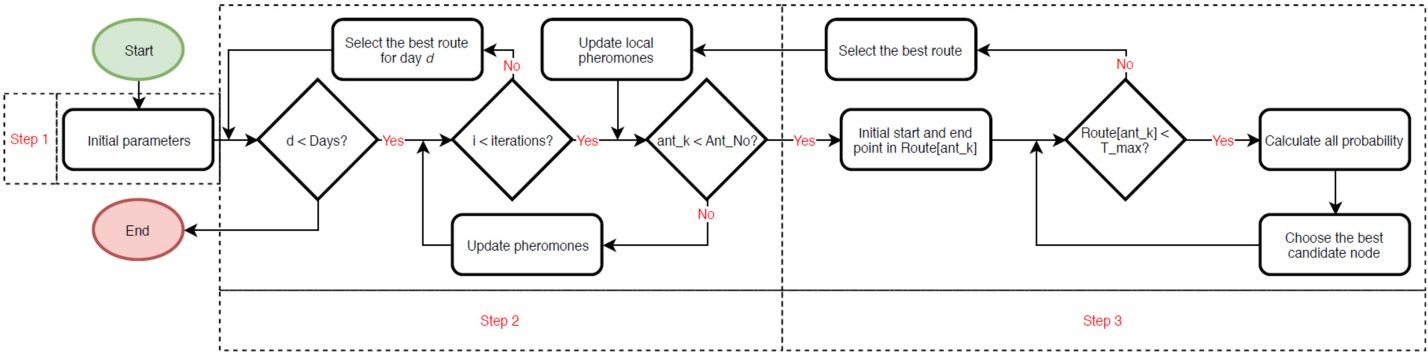

**Figure 2** Flowchart of the ant colony optimization (ACO) algorithm.               

Equation (13) computes the connection value ($C_{ij}$) relative to distance ($TT_{ij}$) and rate of activity value ($S_i$) relative to distance ($TT_{ij}$). The parameter *Eta* signifies the user's preference for the POI. Furthermore, Equation 14 delineates the probability computation for transitioning from POI *i* to *j*. Equation 15 illustrates the function that determines the maximum total score achieved by $Ant_x$, while Equation 16 represents the local update (the initial update). Additionally, Equation 17 depicts the global update (the subsequent update).

$$\eta_{ij} = \frac{S_i}{TT_{ij}} + \frac{C_{ij}}{TT_{ij}} \tag{13}$$

$$P_{i,j} = \frac{(\tau_{i,j})^\alpha (\eta_{ij})^\beta}{\Sigma \left( (\tau_{i,j})^\alpha (\eta_{ij})^\beta \right)} \tag{14}$$

$$\delta_{i,j} = Max(\delta_{i,j}, Ant_x(i,j)) \tag{15}$$

$$\tau_{i,j} = (1 - \rho) \times \tau_{i,j} + \delta_{i,j} \tag{16}$$

$$\tau_{i,j} = \rho \times \tau_{i,j} + (1 - \rho) \times \delta_{i,j} \tag{17}$$

**Overview of HM**

A comprehensive overview of the HM is presented in this section by illustrating its design to address certain limitations inherent in existing methodologies. Primarily, in classifying the diverse actions associated with trips (such as activities, connections, and waiting periods), time emerges as a critical component within the HM framework. The HM aims to optimize the aggregate scores accrued from these three trip-related actions. assesses if users are involved in an activity, moving between activities, or waiting. Subsequently, the HM calculates the combined scores based on these different actions.

Figure 3 exemplifies the HM's management of the distinct actions involved in a trip. In this illustration, POIs #13 and #25 are associated with multi-scores. The chosen route from the start point to #13, from #13 to #25, and from #25 to the end point is also attributed with

| Algorithm 1 | Happiness model optimization. |
| --- | --- |

**1: Initialization:**

Initialize pheromone levels $\tau_{i,j}$ for all edges $(i, j) \in E$.

Initialize ant positions randomly.

**2. Construction phase:**

**for** each ant **do**

    Construct a solution by iteratively selecting the next node to visit based on a probability distribution that considers the pheromone levels and heuristic information (Eq. (14)).

**end for**

**3. Evaluation phase:**

**for** each ant **do**

    Evaluate the fitness of the ant's solution based on the objective function (Eq. (5)).

**end for**

**4. Pheromone update:**

**for** each edge $(i, j) \in E$ **do**

    Update pheromone level $\tau_{i,j}$ based on the quality of the solutions found (Eq. (15)).

**end for**

**5. Repeat steps 2–4 until a termination criterion is met.**

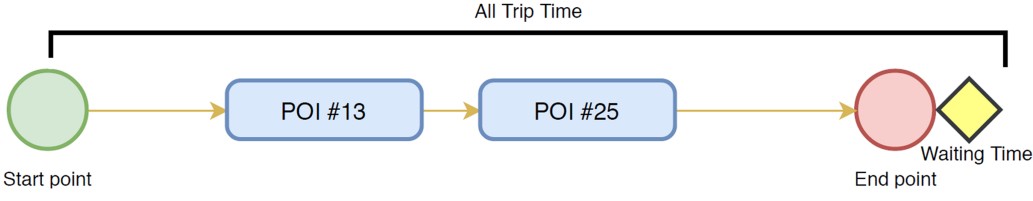

**Figure 3 An example for the calculation of the happiness model.**

multiple scores. Waiting time is depicted as a yellow diamond, indicating periods where travelers have additional time for extra activities, even though the recommendation systems propose an itinerary shorter than the maximum available time ($T_{max}$). Furthermore, the value of each action is scaled by the duration required for its completion (for instance, the transfer from POI#13 to POI#25 spans 19 min, thus calculated as 19 min $\times Score_{13,25}$).

## Overview of HM with ICDM

This section elucidates the integration of the ICDM with the HM. In essence, the ICDM is engineered to address multi-item constraints, whereas the HM is tailored to manage multiple values for POIs and connections. Figure 4 provides an overview of the HM and illustrates the collaborative functioning of the HM and ICDM. This figure are segmented into three main components: (1) data, (2) the HM, and (3) the ICDM.

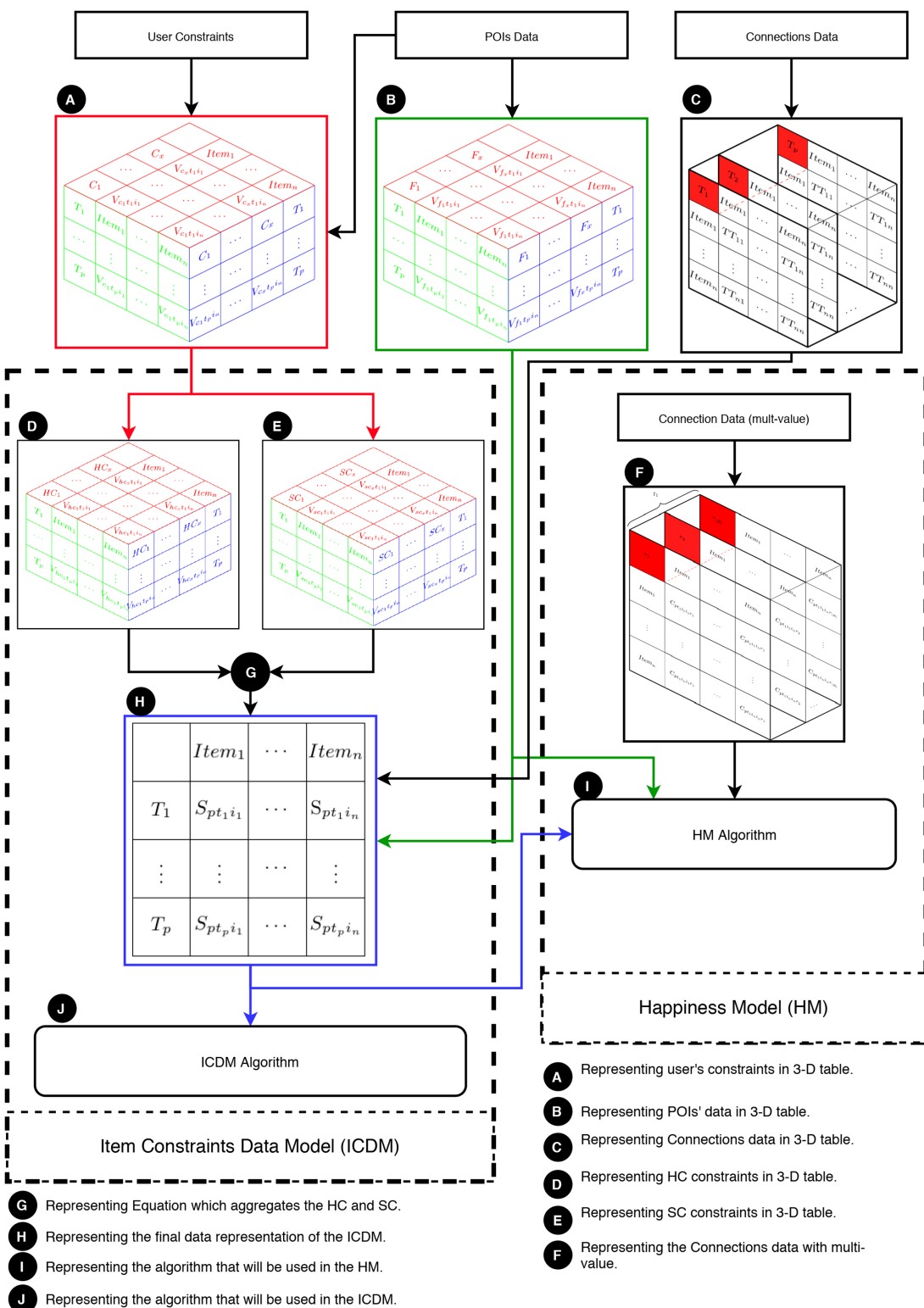

**Figure 4 Overview of the HM and its integration with the ICDM.**

Primarily, the ICDM addresses POIs data, connections data, and user constraints. Essentially, the user constraints are conditions applied to some or all POIs, considering factors like location, entrance fee, or weather conditions. These constraints are presented as a three-dimensional table (labeled as A in Fig. 4). The primary aspects of user constraints are: (1) each constraint assigns a value to an item at a specific moment; (2) at each moment, each constraint assigns a value to some items; and (3) each item is associated with some constraints at every moment. The POIs data contains information about the POIs, including their opening/closing times, locations, and categories, and is displayed in a three-dimensional table (labeled as B in Fig. 4). Moreover, connections data refers to travelers' movements from one POI to another (labeled as C in Fig. 4). The ICDM takes into account congestion levels, which are represented as layers of $p$ tables, where each table indicates the time consumed in traveling from one POI to another at time $T_p$.

Secondly, the core of the ICDM lies in its constraints, categorized into hard constraints (HC) and soft constraints (SC): (1) HC (denoted as D in Fig. 4) and (2) SC (denoted as E in Fig. 4). Equation (4) (denoted as G in Fig. 4) aggregates all constraints values into a single value ($S_{pti}$). The primary output of the model is a matrix that narrows down the search space (denoted as H in Fig. 4).

Thirdly, the HM is designed to address user constraints utilizing the ICDM while taking into account various connection values (represented as F in Fig. 4). The subsequent graph demonstrates the intricate nature of connection data when multiple values are included. Fig. 5 displays $|R|$ tables based on the quantity of multi-values in the connections at a specific moment, such as $t_1$. It requires a set of aggregated tables, denoted as $p$, to comprehensively address the entire multi-value issue. Finally, an algorithm, denoted as I in Fig. 4, is devised to solve the HM problem.

**Remark 2.** *As compared to our previously published study (Alatiyyah, 2024) where a flexible travel recommender model (FTRM) was proposed, the happiness model proposed in this study significantly advances the concepts established by the FTRM model by shifting the focus from symmetry between user preferences and travel constraints to maximizing traveler satisfaction throughout the trip. Unlike the FTRM's POI centric approach, the HM employs a novel time-centric framework for itinerary customization, optimizing user happiness dynamically over time. Additionally, the HM incorporates a broader range of activities, including travel between POIs and minimizing wasted time, allowing for a more personalized and efficient travel experience. It also addresses data dimensionality and search space size in conjunction with the item constraints data model (ICDM), facilitating quicker recommendations tailored to individual needs. Thus, the proposed HM represents a significant evolution in travel recommendation systems, enhancing personalization and satisfaction compared to the FTRM.*

# EXPERIMENTS

## Background of the experiment

The experiments were designed to evaluate the effectiveness of the proposed HM in generating personalized tour recommendations. The primary goal was to demonstrate how the HM optimizes traveler satisfaction by considering activities, connections, and waiting

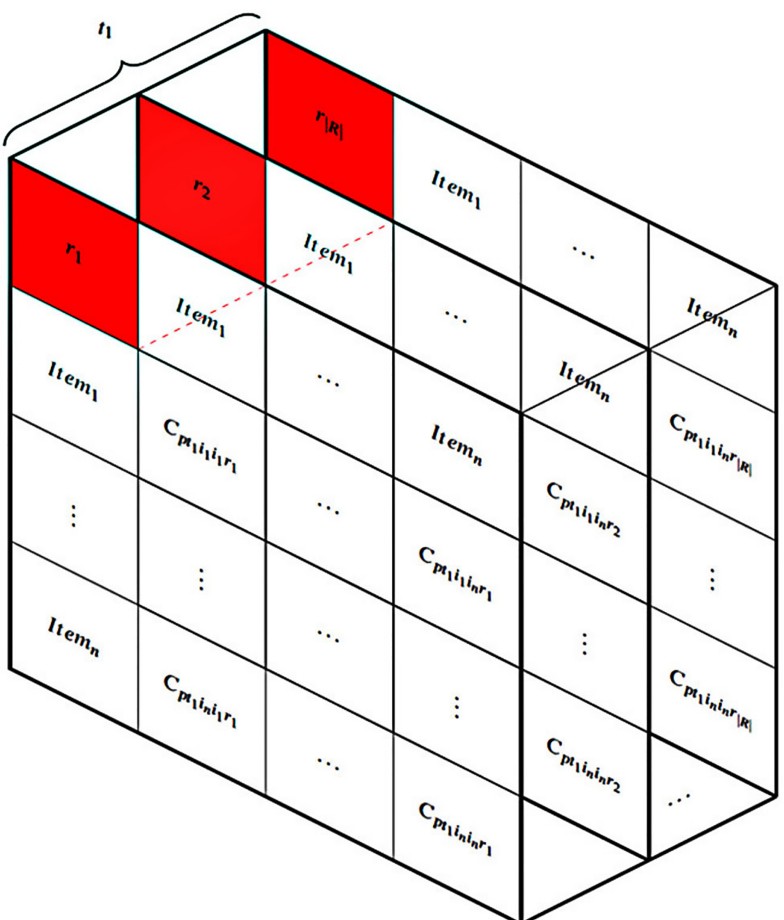

**Figure 5  Multi-value connections in the HM.**     

times, addressing the limitations of existing models. The experiments were conducted using publicly available datasets, including the OPTW and TOPTW datasets, which provide features such as time windows, visiting times, and POI attributes.

**Remark 3.** *OPTW and TOPTW are standard datasets which are widely used in the research community to simulate realistic travel scenarios, allowing for fair comparisons with existing models. Moreover, such datasets provide a controlled environment to evaluate the HM's core functionality, such as optimizing activities, connections, and waiting times. Furthermore, these datasets include a large number of POIs and scenarios, enabling us to test the HM's scalability and efficiency.*

## Method of the experiment

The experiments were conducted in two phases:

- Dataset preparation: The OPTW and TOPTW datasets were preprocessed to align with the HM's requirements. This included extracting POI attributes, travel times, and user constraints.

- Model evaluation: The HM was implemented using the ACO algorithm to solve the optimization problem. The performance of the HM was evaluated based on satisfaction scores, waiting times, and the number of recommended POIs.

## The eight experiments E1, E2,…, E8

To comprehensively evaluate the HM, eight experiments (E1 to E8) were conducted, each with different configurations of connection and waiting time preferences. These experiments were designed to test the HM's ability to adapt to varying user preferences and constraints. The details of each experiment are as follows:

- E1: Connection score = 1, Waiting time score = 0.
- E2: Connection score = 1, Waiting time score = 0.5.
- E3: Connection score = Random, Waiting time score = 0.
- E4: Connection score = Random/2, Waiting time score = 0.
- E5: Connection score = 1, Waiting time score = 0 (applied to Dataset 2).
- E6: Connection score = 1, Waiting time score = 0.5 (applied to Dataset 2).
- E7: Connection score = Random, Waiting time score = 0 (applied to Dataset 2).
- E8: Connection score = Random/2, Waiting time score = 0 (applied to Dataset 2).

These experiments allowed us to analyze how different configurations of connection and waiting time preferences impact the HM's performance.

## Features of the original dataset

The experiments were conducted using two datasets:

- Dataset 1: Derived from the OPTW dataset, this dataset includes 48 to 288 POIs across 10 scenarios. Each POI has attributes such as visiting time, time windows, and satisfaction scores.
- Dataset 2: Derived from the TOPTW dataset, this dataset also includes 48 to 288 POIs across 10 scenarios, with additional features such as travel times between POIs and user constraints.

In our investigation, multiple experiments are conducted to evaluate the capabilities of the HM model. At the outset, the HM model is applied to various pre-existing datasets (refer to Table 2) to assess its effectiveness in constructing tour itineraries. Additionally, an ACO algorithm is devised to generate results that are comparable to those produced by existing models.

The datasets were selected based on the availability of public data, with a specific emphasis on the OPTW and TOPTW datasets (*KU Leuven, 2024*). These datasets were chosen for the HM experiment due to their provision of several pertinent features, such as time windows and visiting times.

Multiple experiments are carried out to find the optimal ACO parameters. The initial parameters for ACO are detailed in Table 3, and the variations in values based on different parameter settings are illustrated in Figs. 6 and 7.

**Table 2  Datasets used in experiments.**

| Problem | Dataset | Study | # Scenarios | # items $|I|$ |
|---|---|---|---|---|
| OPTW & TOPTW | Dataset 1 | *Vansteenwegen et al. (2009)* | 10 | 48 to 288 |
| | Dataset 2 | | 10 | 48 to 288 |

**Table 3  The parameters of the algorithm in the initial step.**

| Parameter | Value | Representation |
|---|---|---|
| $\alpha$ | 15 | Represents the importance of *Tau* |
| $\beta$ | 8 | Represents the significance of *Eta* |
| $\rho$ | 0.1 | Value of pheromone evaporation |
| *Ant_No* | 200 | # ants |
| *Iterations* | 10 | # iteration |
| *NodeSize* | | # nodes |
| $\eta_{ij}$ | Equation (13) | *Eta* Indicates the rate of score to distance |
| $\tau_{i,j}$ | Allocate 1,000 value | Representing the level of Pheromones from $i$ to $j$ |
| $\delta_{i,j}$ | Allocate 0 value | Representing the maximum total path use $i$ to $j$ |

A series of experiments are conducted to determine the best iteration number for the algorithm. It is critical to recognize that the running time is a pivotal factor, particularly when dealing with NP-hard problems. Figure 8 illustrates the average running time of the ACO algorithm for different numbers of iterations. The results demonstrate that the running time increases linearly with the number of iterations, ranging from 2.9 s for 10 iterations to 26.7 s for 100 iterations. This linear relationship highlights the scalability of the ACO algorithm, making it suitable for solving the Happiness Model's optimization problem efficiently. For experimentation, ten different values are selected for the iteration parameter.

*Alpha* and *Beta* denote the significance of the score and the rate of score relative to the distance (refer to Equation (14)). Selecting the best values for such parameters is crucial as the performance of the algorithm hinges on these parameters. Thus, a tuning method is employed that iteratively adjusts these parameters from zero to the point of maximum gain. Multiple experiments were conducted (each scenario within each dataset was executed over 200 times for different values of the parameters) to identify the optimal performance of the algorithm across all datasets. Figures 6 and 7 illustrate the normalized total scores (where 1 represents the highest total scores across all datasets) for various *Alpha* and *Beta* values in all scenarios.

As depicted in Figs. 6 and 7, only a subset of the results attained top scores, while the others varied; no single pair of *Alpha* and *Beta* values consistently yielded uniform algorithm performance (each distinct value for *Alpha* and *Beta* produced variant outcomes). The best values for these parameters are found to be *Alpha* = 1 and *Beta* = 13,

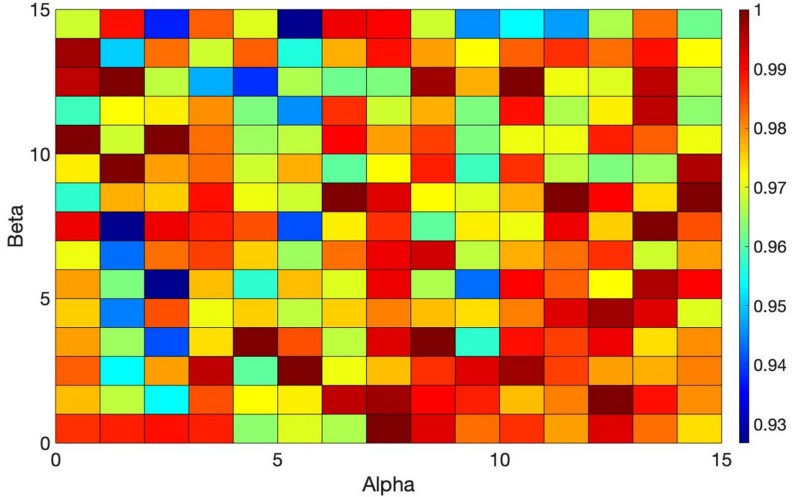

**Figure 6 Overview of the ACO's performance based on different values of *Alpha* and *Beta*.**

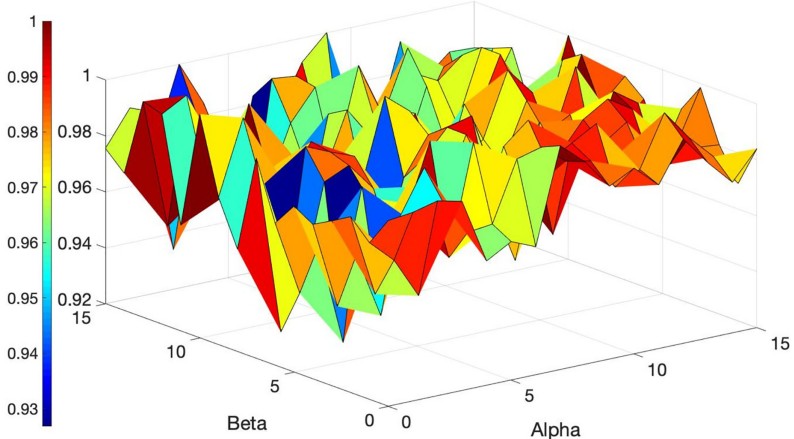

**Figure 7 Overview of the ACO's performance based on different values of *Alpha* and *Beta*.**

*Alpha* = 15 and *Beta* = 8, and others, that consistently delivered excellent performance across all datasets. You can see these values highlighted in dark red in Figs. 6 and 7 which achieved the best results. Essentially, any values in dark red will consistently produce good results across all datasets.

## RESULTS

The outcomes obtained by the HM through experimental analysis are presented in this section. Furthermore, the discussion elucidates the inherent challenges and distinctive characteristics of the extant datasets.

### The HM benchmark instances

The outcomes of the proposed HM with those derived from the TOPTW on Dataset 1 and Dataset 2 are presented in this section (additional details are provided in Table 2). Broadly,

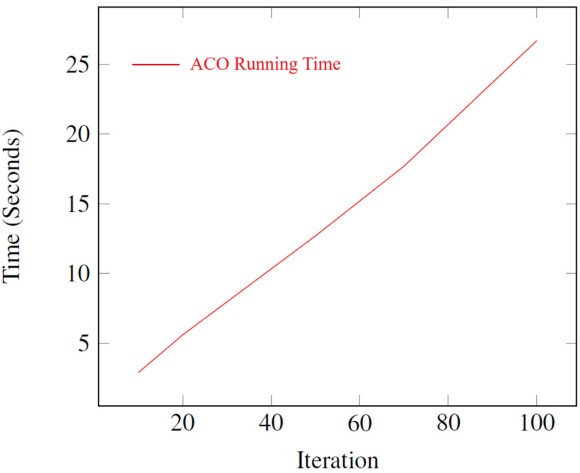

**Figure 8 Average running time for different iteration numbers in the ACO algorithm.**

the findings elucidate the influence of the HM in structuring a tour trip by tailoring the itinerary according to user preferences. Specifically, the HM results indicate that a higher degree of personalization in the trips is attainable, as the model integrates and optimizes additional elements of trip itineraries like idle time and connectivity. Furthermore, the metrics for activities score, waiting times and connections have been standardized to a scale ranging from zero to one, with one denoting the maximal value.

Table 4 enumerates the 8 distinct experiments carried on the model. Initially, the values for activities were based on the public dataset, following which various values for connections and waiting times were generated in these experiments. Such experiments encompass diverse scenarios to underscore the advantages conferred by the model.

Experiments $E_1$ to $E_4$ were conducted on Dataset 1, and the outcomes of these experiments are detailed in Tables 5–8. Each experiment provides the code for the scenario along with the number of POIs included in the recommended trip. Additionally, the primary three activities of the trip are identified, with the corresponding durations for each activity and the total scores assigned to each specific activity presented.

Table 5 presents the outcomes of $E_1$, as specified by the parameters outlined in Table 4. Each entry in the table corresponds to a distinct scenario in which the cumulative scores for each action vary across scenarios, with the exception of the total waiting-time scores, which are consistently zero (refer to Table 4). Furthermore, all action scores are normalized to a range between 0 and 1, ensuring that the total score represents the sum of individual moments, and the total score for any action is never greater than the total number of moments in the activity.

The results of this table demonstrate the HM's ability to generate itineraries with high activity scores and efficient connections. For example, in scenario Pr02, the HM recommends 19 POIs with a total activity score of 125 and a connection score of 384. The absence of waiting time scores indicates that the HM prioritizes minimizing idle time, aligning with the goal of maximizing traveler satisfaction.

**Table 4 Comparison results.**

| Experiment | Connection | Waiting | Dataset |
|---|---|---|---|
| $E_1$ | 1 | 0 | Dataset 1 |
| $E_2$ | 1 | 0.5 | Dataset 1 |
| $E_3$ | Random | 0 | Dataset 1 |
| $E_4$ | $\frac{Random}{2}$ | 0 | Dataset 1 |
| $E_5$ | 1 | 0 | Dataset 2 |
| $E_6$ | 1 | 0.5 | Dataset 2 |
| $E_7$ | Random | 0 | Dataset 2 |
| | $\frac{Random}{2}$ | 0 | Dataset 2 |

**Table 5 Result of $E_1$ on the happiness model.**

| Code | Activity | | Connection | | Waiting | | # POIs |
|---|---|---|---|---|---|---|---|
| | Time (M) | Scores | Time (M) | Scores | Time (M) | Scores | |
| Pr02 | 227 | 125 | 384 | 384 | 389 | 0 | 19 |
| Pr03 | 197 | 93 | 368 | 368 | 435 | 0 | 17 |
| Pr04 | 223 | 120 | 382 | 382 | 395 | 0 | 19 |
| Pr05 | 389 | 245 | 351 | 351 | 260 | 0 | 24 |
| Pr06 | 304 | 145 | 371 | 371 | 325 | 0 | 24 |
| Pr08 | 252 | 120 | 417 | 417 | 331 | 0 | 18 |
| Pr09 | 222 | 120 | 464 | 464 | 314 | 0 | 20 |
| Pr10 | 276 | 160 | 403 | 403 | 321 | 0 | 25 |

**Table 6 Result of $E_2$ on the happiness model.**

| Code | Activity | | Connection | | Waiting | | # POIs |
|---|---|---|---|---|---|---|---|
| | Time (M) | Scores | Time (M) | Scores | Time (M) | Scores | |
| Pr02 | 227 | 125 | 384 | 384 | 389 | 194 | 19 |
| Pr03 | 197 | 93 | 368 | 368 | 435 | 217 | 17 |
| Pr04 | 223 | 120 | 382 | 382 | 395 | 197 | 19 |
| Pr05 | 389 | 245 | 351 | 351 | 260 | 130 | 24 |
| Pr06 | 304 | 145 | 371 | 371 | 325 | 162 | 24 |
| Pr08 | 252 | 120 | 417 | 417 | 331 | 166 | 18 |
| Pr09 | 222 | 120 | 464 | 464 | 314 | 157 | 20 |
| Pr10 | 276 | 160 | 403 | 403 | 321 | 160 | 25 |

Table 6 shows the outcomes of Experiment 2 ($E_2$), where the connection score remains 1, but the waiting time score is set to 0.5. The results are similar to E1, with the addition of waiting time scores. For instance, in scenario Pr02, the waiting time score is 194, indicating that the HM can incorporate waiting times when necessary. This flexibility allows the HM

Table 7 Result of $E_3$ on the happiness model.

| Code | Activity | | Connection | | Waiting | | # POIs |
|------|----------|--------|------------|--------|----------|--------|--------|
| | Time (M) | Scores | Time (M) | Scores | Time (M) | Scores | |
| Pr02 | 227 | 125 | 413 | 214 | 360 | 0 | 19 |
| Pr03 | 222 | 110 | 436 | 255 | 342 | 0 | 18 |
| Pr04 | 220 | 117 | 396 | 292 | 384 | 0 | 18 |
| Pr05 | 376 | 234 | 365 | 281 | 259 | 0 | 23 |
| Pr06 | 298 | 141 | 468 | 294 | 234 | 0 | 23 |
| Pr08 | 283 | 139 | 454 | 246 | 263 | 0 | 20 |
| Pr09 | 268 | 150 | 415 | 295 | 317 | 0 | 23 |
| Pr10 | 254 | 145 | 374 | 252 | 372 | 0 | 23 |

Table 8 Result of $E_4$ on the happiness model.

| Code | Activity | | Connection | | Waiting | | # POIs |
|------|----------|--------|------------|--------|----------|--------|--------|
| | Time (M) | Scores | Time (M) | Scores | Time (M) | Scores | |
| Pr02 | 249 | 146 | 414 | 110 | 337 | 0 | 21 |
| Pr03 | 197 | 93 | 443 | 110 | 360 | 0 | 17 |
| Pr04 | 220 | 117 | 366 | 145 | 414 | 0 | 18 |
| Pr05 | 376 | 234 | 367 | 108 | 257 | 0 | 23 |
| Pr06 | 270 | 128 | 393 | 135 | 337 | 0 | 21 |
| Pr08 | 252 | 120 | 446 | 133 | 302 | 0 | 18 |
| Pr09 | 248 | 130 | 420 | 123 | 332 | 0 | 22 |
| Pr10 | 254 | 145 | 457 | 143 | 289 | 0 | 23 |

to adapt to varying user preferences, addressing the limitations of traditional models that treat waiting times as fixed costs.

Table 7 presents the outcomes of Experiment 3 ($E_3$), where the connection score is randomly assigned, and the waiting time score is set to 0. The results highlight the HM's ability to handle varying connection preferences. For example, in scenario Pr02, the connection score drops to 214, but the HM still recommends 19 POIs, demonstrating its robustness in optimizing itineraries under different conditions.

Table 8 shows the outcomes of Experiment 4 ($E_4$), where the connection score is halved, and the waiting time score is set to 0. The results indicate that the HM can generate efficient itineraries even with reduced connection scores. For instance, in scenario Pr02, the connection score is 110, but the HM still recommends 21 POIs, showcasing its ability to balance multiple factors effectively.

Table 9 presents the outcomes of Experiment 5 ($E_5$), applied to Dataset 2. The results demonstrate the HM's scalability and effectiveness across different datasets. For example, in scenario Pr11, the HM recommends 19 POIs with a total activity score of 87 and a

**Table 9 Result of $E_5$ on the happiness model.**

| Code | Activity | | Connection | | Waiting | | #POIs |
|------|----------|--------|------------|--------|---------|--------|-------|
| | Time (M) | Scores | Time (M) | Scores | Time (M) | Scores | |
| Pr11 | 205 | 87 | 394 | 394 | 401 | 0 | 19 |
| Pr12 | 82 | 41 | 201 | 201 | 717 | 0 | 7 |
| Pr13 | 273 | 130 | 347 | 347 | 380 | 0 | 22 |
| Pr14 | 309 | 173 | 259 | 259 | 432 | 0 | 24 |
| Pr15 | 429 | 273 | 327 | 327 | 244 | 0 | 30 |
| Pr16 | 324 | 162 | 345 | 345 | 331 | 0 | 26 |
| Pr17 | 231 | 102 | 379 | 379 | 390 | 0 | 18 |
| Pr18 | 287 | 141 | 350 | 350 | 363 | 0 | 21 |
| Pr19 | 288 | 151 | 351 | 351 | 361 | 0 | 25 |
| Pr20 | 324 | 184 | 313 | 313 | 363 | 0 | 29 |

connection score of 394, highlighting its ability to handle larger and more complex datasets.

Table 10 shows the outcomes of Experiment 6 ($E_6$), applied to Dataset 2. The results are similar to E5, with the addition of waiting time scores. For instance, in scenario Pr11, the waiting time score is 200, indicating that the HM can incorporate waiting times even in larger datasets.

Table 11 presents the outcomes of Experiment 7 ($E_7$), applied to Dataset 2. The results highlight the HM's ability to adapt to random connection scores. For example, in scenario Pr12, the HM recommends 5 POIs with a total activity score of 26 and a connection score of 169, demonstrating its flexibility in handling diverse user preferences.

Table 12 shows the outcomes of Experiment 8 ($E_8$), applied to Dataset 2. The results indicate that the HM can generate efficient itineraries even with reduced connection scores. For instance, in scenario Pr12, the connection score is 70, but the HM still recommends 5 POIs, showcasing its ability to balance multiple factors effectively.

In this study, several simulated user scenarios have been conducted with diverse travel preferences and experiences. The personalized itineraries generated by the HM for these scenarios are checked based on the following aspects: How well did the itinerary align with the provided preferences and interests? How well did the itinerary optimize the time and resources and satisfy the overall travel experience? In fact, the obtained results were overwhelmingly positive. The proposed HM generated highly relevant and personalized itineraries that met the expectations. In addition, the proposed model has the ability to optimize the time and resources, leading to a more efficient and enjoyable travel experience. While the simulated user feedback provides valuable insights, it is important to note that it does not replace real-world user studies. Further research is needed to gather feedback from a larger and more diverse group of users to fully validate the HM's effectiveness in enhancing user satisfaction.

**Table 10 Result of $E_6$ on the happiness model.**

| Code | Activity | | Connection | | Waiting | | #POIs |
|---|---|---|---|---|---|---|---|
| | Time (M) | Scores | Time (M) | Scores | Time (M) | Scores | |
| Pr11 | 205 | 87 | 394 | 394 | 401 | 200 | 19 |
| Pr12 | 82 | 41 | 201 | 201 | 717 | 359 | 7 |
| Pr13 | 273 | 130 | 347 | 347 | 380 | 190 | 22 |
| Pr14 | 309 | 173 | 259 | 259 | 432 | 216 | 24 |
| Pr15 | 429 | 273 | 327 | 327 | 244 | 122 | 30 |
| Pr16 | 324 | 162 | 345 | 345 | 331 | 166 | 26 |
| Pr17 | 231 | 102 | 379 | 379 | 390 | 195 | 18 |
| Pr18 | 287 | 141 | 350 | 350 | 363 | 182 | 21 |
| Pr19 | 288 | 151 | 351 | 351 | 361 | 180 | 25 |
| Pr20 | 324 | 184 | 313 | 313 | 363 | 182 | 29 |

**Table 11 Result of $E_7$ on the happiness model.**

| Code | Activity | | Connection | | Waiting | | #POIs |
|---|---|---|---|---|---|---|---|
| | Time (M) | Scores | Time (M) | Scores | Time (M) | Scores | |
| Pr11 | 182 | 79 | 395 | 200 | 423 | 0 | 17 |
| Pr12 | 51 | 26 | 268 | 169 | 681 | 0 | 5 |
| Pr13 | 266 | 127 | 341 | 209 | 393 | 0 | 21 |
| Pr14 | 292 | 160 | 340 | 241 | 368 | 0 | 23 |
| Pr15 | 406 | 259 | 315 | 183 | 279 | 0 | 28 |
| Pr16 | 298 | 141 | 377 | 227 | 325 | 0 | 23 |
| Pr17 | 237 | 103 | 364 | 236 | 399 | 0 | 19 |
| Pr18 | 290 | 144 | 330 | 216 | 380 | 0 | 22 |
| Pr19 | 306 | 164 | 343 | 209 | 351 | 0 | 26 |
| Pr20 | 324 | 184 | 351 | 205 | 325 | 0 | 29 |

**Remark 4.** *The ACO algorithm has a time complexity of $O(N.M.K)$, where N is the number of ants, M is the number of iterations, and K is the number of POIs. On the other hand, the space complexity is $O(K^2)$, primarily due to the storage of pheromone levels and heuristic values for all edges.*

## Summary of key findings

The results of experiments, presented in Tables 5–12, demonstrate that the proposed Happiness Model effectively addresses the limitations of existing travel recommendation systems by:

- Maximizing traveler satisfaction: The HM generates itineraries with high activity and connection scores, ensuring a satisfying travel experience.

Table 12 Result of $E_8$ on the happiness model.

| Code | Activity | | Connection | | Waiting | | #POIs |
|------|----------|--------|------------|--------|---------|--------|-------|
| | Time (M) | Scores | Time (M) | Scores | Time (M) | Scores | |
| Pr11 | 195 | 81 | 355 | 94 | 450 | 0 | 18 |
| Pr12 | 51 | 26 | 221 | 70 | 728 | 0 | 5 |
| Pr13 | 266 | 127 | 371 | 113 | 363 | 0 | 21 |
| Pr14 | 309 | 173 | 312 | 78 | 379 | 0 | 24 |
| Pr15 | 397 | 251 | 331 | 112 | 272 | 0 | 27 |
| Pr16 | 308 | 148 | 382 | 110 | 310 | 0 | 25 |
| Pr17 | 231 | 102 | 401 | 111 | 368 | 0 | 18 |
| Pr18 | 283 | 139 | 293 | 81 | 424 | 0 | 20 |
| Pr19 | 306 | 164 | 363 | 96 | 331 | 0 | 26 |
| Pr20 | 328 | 184 | 329 | 81 | 343 | 0 | 30 |

- Minimizing waiting times: The HM incorporates waiting times as a key factor, reducing idle time and improving efficiency.
- Adapting to user preferences: The HM can handle varying connection and waiting time preferences, showcasing its flexibility and robustness.
- Scaling to larger datasets: The HM performs well on both Dataset 1 and Dataset 2, demonstrating its scalability and applicability to real-world scenarios.

While many existing models, such as MOOP and MCTOPMTW, focus primarily on optimizing the selection of POIs based on specific dimensions (*e.g.*, total scores, distance), the proposed model takes a broader approach. The proposed HM considers a wider range of factors—such as multi-value (MV) connections, waiting times, and personalized travel constraints—that are often overlooked in existing techniques. Thus, comparing these models directly might be misleading as they address different aspects of travel planning.

Despite the differences in factors studied, the proposed HM model offers several distinct advantages over existing approaches:

- *Multi-value (MV) consideration for nodes and connections:* Unlike models like MOOP, which evaluate POIs based solely on scores or distances, HM incorporates multiple attributes for each POI and the transitions between them (*e.g.*, time, cost, journey length). This multi-dimensional evaluation provides a more comprehensive understanding of traveler satisfaction.
- *Aggregation for traveler satisfaction:* While MOOP focuses on maximizing benefits within POI categories, HM aggregates multiple values into a single metric representing the traveler's overall satisfaction, factoring in elements such as time spent at POIs, journey length, and costs.
- *Personalization and waiting time consideration:* One key limitation of models like MOOP is the lack of support for personalization, such as waiting time, which is

crucial for travelers with reservations (*e.g.*, flights, hotels). HM addresses this by including waiting time as an essential factor, enhancing its suitability for real-world travel scenarios.

- *Handling multiple decision-making parameters:* Unlike the more limited scope of MOOP and MCTOPMTW, HM covers a comprehensive range of tourist decision-making parameters by categorizing them into: Activities, Connections, and Waiting Time.

This broader set of factors makes the proposed HM uniquely equipped to handle the complexities of modern travel planning, unlike MOOP, which considers only multi-value POIs without addressing connections or waiting time.

It is worth mentioning that the proposed HM differs from the existing models not only in its handling of multiple values for POIs but also in its inclusion of transitions between POIs and waiting time considerations. In contrast:

- *HM* provides a comprehensive approach by integrating multi-value POIs, multi-value connections, and waiting time, offering a more comprehensive solution for travel planning.
- *MOOP* lacks personalization and does not consider waiting time or multi-value transitions, focusing only on maximizing benefits within POI categories.
- *MCTOPMTW* supports multiple constraints but lacks an effective aggregation mechanism like HM's traveler satisfaction metric, making it more complex and less intuitive.

Although existing models such as MOOP and MCTOPMTW have certain strengths, they are limited in scope. The proposed model provides a more comprehensive and personalized approach by incorporating multi-value POIs, connections, and waiting time, making it better suited to meet the needs of travelers in practical, real-world scenarios.

## DISCUSSION

This section elaborates on the findings derived from the HM. It is important to note that since no waiting option is provided for travelers between points of interest (POIs), adjustments to the weighting of waiting time are unlikely to influence the outcomes.

In experiments $E_2$ and $E_6$, each instance of waiting has been assigned a value of 0.5. Nevertheless, due to the algorithm's omission of the waiting option, the outcomes are primarily influenced by the activity and connection values. The trajectories of $E_2$ and $E_6$ are identical to those in $E_1$ and $E_5$, with the primary distinction being the value assigned to waiting time.

Tables 13 and 14 present a comparative analysis of varying preference values for waiting times. Specifically, in scenarios $E_1$ and $E_5$, the waiting time is set to 0, whereas in scenarios $E_2$ and $E_6$, the waiting time is set to 0.5. The primary conclusion drawn from these comparisons is that differing waiting-time values do not influence the recommended itinerary. This outcome is attributed to the assumption that travelers do not desire the

**Table 13  Comparison results $E_1$ and $E_2$.**

| Experiment | Code | Activity | | Connection | | Waiting | | # POIs |
|---|---|---|---|---|---|---|---|---|
| | | Time (M) | Score | Time (M) | Score | Time (M) | Score | |
| $E_1$ | Pr02 | 227 | 125 | 384 | 384 | 389 | 0 | 19 |
| $E_2$ | | 227 | 125 | 384 | 384 | 389 | 194 | 19 |
| $E_1$ | Pr03 | 197 | 93 | 368 | 368 | 435 | 0 | 17 |
| $E_2$ | | 197 | 93 | 368 | 368 | 435 | 217 | 17 |
| $E_1$ | Pr04 | 223 | 120 | 382 | 382 | 395 | 0 | 19 |
| $E_2$ | | 223 | 120 | 382 | 382 | 395 | 197 | 19 |
| $E_1$ | Pr05 | 389 | 245 | 351 | 351 | 260 | 0 | 24 |
| $E_2$ | | 389 | 245 | 351 | 351 | 260 | 130 | 24 |
| $E_1$ | Pr06 | 304 | 145 | 371 | 371 | 325 | 0 | 24 |
| $E_2$ | | 304 | 145 | 371 | 371 | 325 | 162 | 24 |
| $E_1$ | Pr08 | 252 | 120 | 417 | 417 | 331 | 0 | 18 |
| $E_2$ | | 252 | 120 | 417 | 417 | 331 | 166 | 18 |
| $E_1$ | Pr09 | 222 | 120 | 464 | 464 | 314 | 0 | 20 |
| $E_2$ | | 222 | 120 | 464 | 464 | 314 | 157 | 20 |
| $E_1$ | Pr10 | 276 | 160 | 403 | 403 | 321 | 0 | 25 |
| $E_2$ | | 276 | 160 | 403 | 403 | 321 | 160 | 25 |

option of waiting between activities. In other words, the model does not incorporate waiting times between points of interest (POIs) for travelers. Nevertheless, the overall waiting-time values fluctuate with varying waiting-time preferences. In conclusion, the assumption that travelers do not prefer to wait between POIs implies that variations in waiting-time preferences do not impact the suggested tour itinerary.

Table 15 presents a comparative analysis of the outcomes from various experimental scenarios. Initially, it is observed that Experiment $E_1$ includes at least 50% more POIs in the recommended tour outcomes. This is attributed to the uniformity of connection values, with all edges between POIs being assigned a value of 1, indicative of the highest satisfaction level. Furthermore, the analysis reveals that Experiments $E_3$ and $E_4$ outperform $E_1$ with respect to minimizing waiting times. The underlying cause for $E_1$'s relatively lower performance compared to $E_3$ and $E_4$ is its incorporation of the longest paths between POIs, as it accommodates user preferences for extended journeys between POIs. Additionally, $E_4$ generates a tour characterized by a total connection time that is at least 50% longer, as it selects shorter connection periods, thereby enabling the inclusion of more POIs within the model.

Another salient aspect is that the HM is designed to customize tour itineraries in accordance with user preferences. Empirical results indicate that the HM effectively fulfills this objective. Figure 9 illustrates the comparative performance of the HM and the TOPTW using an identical dataset (Dataset 2) and algorithm.

**Table 14 Comparison results $E_5$ and $E_6$.**

| Experiment | Code | Activity | | Connection | | Waiting | | # POIs |
|---|---|---|---|---|---|---|---|---|
| | | Time (M) | Score | Time (M) | Score | Time (M) | Score | |
| $E_5$ | Pr11 | 205 | 87 | 394 | 394 | 401 | 0 | 19 |
| $E_6$ | | 205 | 87 | 394 | 394 | 401 | 200 | 19 |
| $E_5$ | Pr12 | 82 | 41 | 201 | 201 | 717 | 0 | 7 |
| $E_6$ | | 82 | 41 | 201 | 201 | 717 | 359 | 7 |
| $E_5$ | Pr13 | 273 | 130 | 347 | 347 | 380 | 0 | 22 |
| $E_6$ | | 273 | 130 | 347 | 347 | 380 | 190 | 22 |
| $E_5$ | Pr14 | 309 | 173 | 259 | 259 | 432 | 0 | 24 |
| $E_6$ | | 309 | 173 | 259 | 259 | 432 | 216 | 24 |
| $E_5$ | Pr15 | 429 | 273 | 327 | 327 | 244 | 0 | 30 |
| $E_6$ | | 429 | 273 | 327 | 327 | 244 | 122 | 30 |
| $E_5$ | Pr16 | 324 | 162 | 345 | 345 | 331 | 0 | 26 |
| $E_6$ | | 324 | 162 | 345 | 345 | 331 | 166 | 26 |
| $E_5$ | Pr17 | 231 | 102 | 379 | 379 | 390 | 0 | 18 |
| $E_6$ | | 231 | 102 | 379 | 379 | 390 | 195 | 18 |
| $E_5$ | Pr18 | 287 | 141 | 350 | 350 | 363 | 0 | 21 |
| $E_6$ | | 287 | 141 | 350 | 350 | 363 | 182 | 21 |
| $E_5$ | Pr19 | 288 | 151 | 351 | 351 | 361 | 0 | 25 |
| $E_6$ | | 288 | 151 | 351 | 351 | 361 | 180 | 25 |
| $E_5$ | Pr20 | 324 | 184 | 313 | 313 | 363 | 0 | 29 |
| $E_6$ | | 324 | 184 | 313 | 313 | 363 | 182 | 29 |

Figure 9 illustrates 4 trips based on several experiments ($E_5$, $E_7$, $E_8$, and TOPTW). The key difference among such experiments is the different preferences for the connections and waiting scores. First, there are no intrinsically 'good' or 'bad' outcomes here (for $E_5$, $E_7$, $E_8$, and TOPTW) since each of these tours is tailored to individual user preferences (one user may enjoy a certain tour while another may not). In general, the primary focus of RSs (personalization) are addressed in which the proposed model assists users in receiving tour recommendations that align with their specific preferences and needs.

Table 16 presents a comparative analysis of the outcomes of experiments $E_5$, $E_7$, $E_8$, and TOPTW. A key observation from the data is that the total number of Points of Interest (POIs) included in each tour: $E_5$ included 7 POIs, in which both $E_7$ and $E_8$ included 5 POIs. Furthermore, the total connection time for $E_5$ is notably lower than that of the other experiments by a margin of at least 10%.

Table 17 delineates the recommended tour outcomes across various experimental conditions. A particularly significant observation is that the results of $E_5$ exhibit substantial deviations from those of the other experiments, notably affecting the majority of the suggested POIs. In contrast, $E_7$, $E_8$, and the TOPTW algorithm demonstrate a notable

**Table 15 Comparing results for all experiments.**

| Code | Experiment | Activity Time (M) | Score | Connection Time (M) | Score | Waiting Time (M) | Score | #POIs |
|------|-----------|---------|-------|---------|-------|---------|-------|-------|
| Pr02 | $E_1$ | 227 | 125 | 384 | 384 | 389 | 0 | 19 |
|      | $E_3$ | 227 | 125 | 413 | 214 | 360 | 0 | 19 |
|      | $E_4$ | 249 | 146 | 414 | 110 | 337 | 0 | 21 |
| Pr03 | $E_1$ | 197 | 93  | 368 | 368 | 435 | 0 | 17 |
|      | $E_3$ | 222 | 110 | 436 | 255 | 342 | 0 | 18 |
|      | $E_4$ | 197 | 93  | 443 | 110 | 360 | 0 | 17 |
| Pr04 | $E_1$ | 223 | 120 | 382 | 382 | 395 | 0 | 19 |
|      | $E_3$ | 220 | 117 | 396 | 292 | 384 | 0 | 18 |
|      | $E_4$ | 220 | 117 | 366 | 145 | 414 | 0 | 18 |
| Pr05 | $E_1$ | 389 | 245 | 351 | 351 | 260 | 0 | 24 |
|      | $E_3$ | 376 | 234 | 365 | 281 | 259 | 0 | 23 |
|      | $E_4$ | 376 | 234 | 367 | 108 | 257 | 0 | 23 |
| Pr06 | $E_1$ | 304 | 145 | 371 | 371 | 325 | 0 | 24 |
|      | $E_3$ | 298 | 141 | 468 | 294 | 234 | 0 | 23 |
|      | $E_4$ | 270 | 128 | 393 | 135 | 337 | 0 | 21 |
| Pr08 | $E_1$ | 252 | 120 | 417 | 417 | 331 | 0 | 18 |
|      | $E_3$ | 283 | 139 | 454 | 246 | 263 | 0 | 20 |
|      | $E_4$ | 252 | 120 | 446 | 133 | 302 | 0 | 18 |
| Pr09 | $E_1$ | 222 | 120 | 464 | 464 | 314 | 0 | 20 |
|      | $E_3$ | 268 | 150 | 415 | 295 | 317 | 0 | 23 |
|      | $E_4$ | 248 | 130 | 420 | 123 | 332 | 0 | 22 |
| Pr10 | $E_1$ | 276 | 160 | 403 | 403 | 321 | 0 | 25 |
|      | $E_3$ | 254 | 145 | 374 | 252 | 372 | 0 | 23 |
|      | $E_4$ | 254 | 145 | 457 | 143 | 289 | 0 | 23 |

degree of convergence, with $E_8$ and TOPTW sharing approximately 66% similarity in their recommended POIs.

Figure 9 provides line graph that visually compares the paths recommended by the proposed happiness model and TOPTW for scenario Pr12 in Dataset 2. The figure highlights the differences in the number of POIs visited, and demonstrates a superior ability for the happiness model to personalize tour itineraries according to user preferences when compared to TOPTW model.

**Remark 5.** *While real-world user data is not currently available for this study, the proposed HM is rigorously tested using publicly available datasets (e.g., OPTW and TOPTW). These datasets include features such as time windows and visiting times, which simulate realistic scenarios. Future studies will focus on validating the model with real-world data and user feedback, ensuring its practical utility in real-life applications*

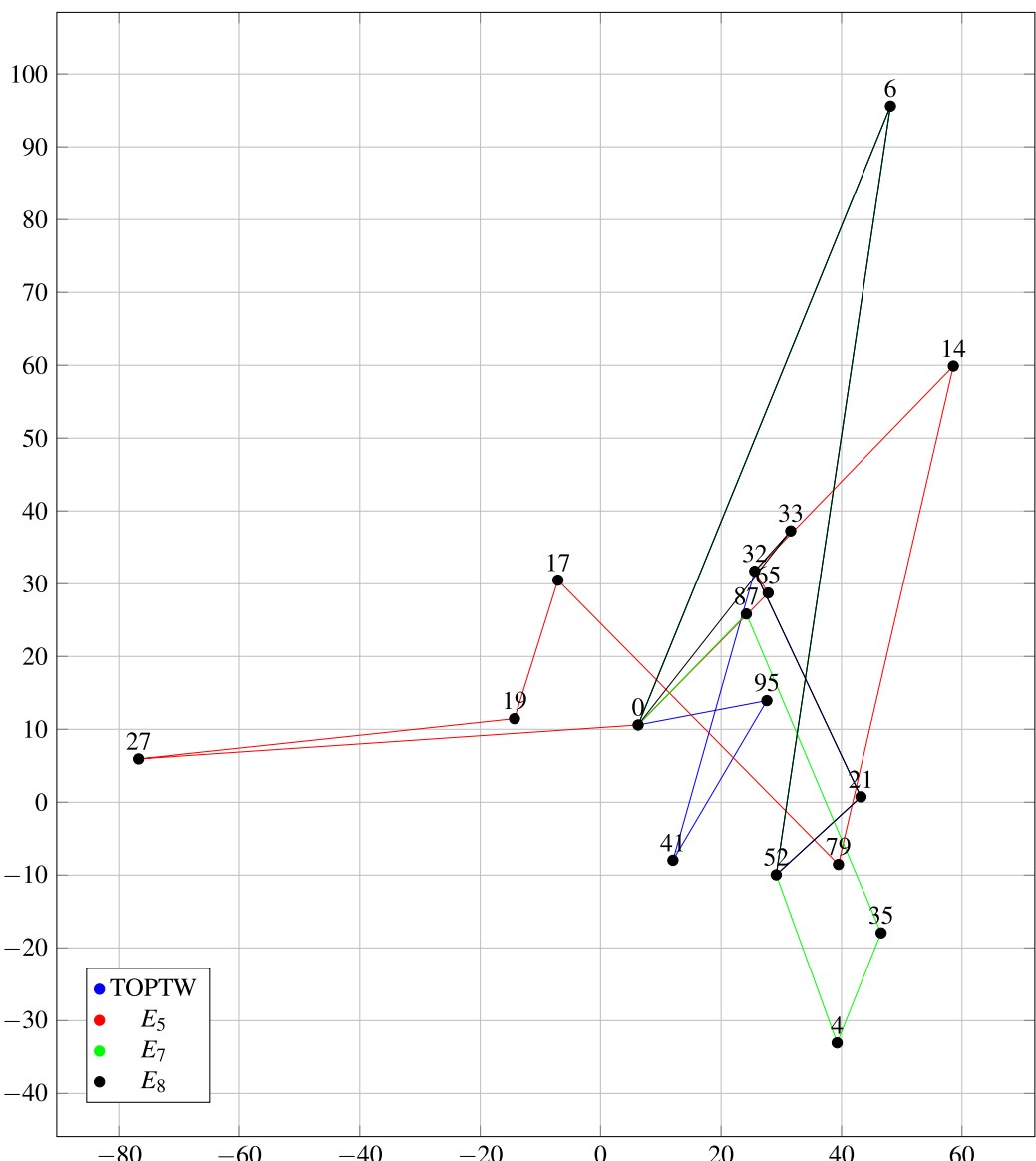

**Figure 9** Comparison of HM and TOPTW paths based on *Pr*12 on Dataset 2.

**Table 16** Comparison results of scenario Pr12 in $E_5$, $E_7$, and $E_8$.

| Experiment | Code | Activity | | Connection | | Waiting | | # POIs |
|---|---|---|---|---|---|---|---|---|
| | | Time (M) | Score | Time (M) | Score | Time (M) | Score | |
| $E_5$ | Pr12 | 82 | 41 | 201 | 201 | 717 | 0 | 7 |
| $E_7$ | Pr12 | 51 | 26 | 268 | 169 | 681 | 0 | 5 |
| $E_8$ | Pr12 | 51 | 26 | 221 | 70 | 728 | 0 | 5 |
| *TOPTW* | Pr12 | 66 | 39 | 320 | 0 | 614 | 0 | 6 |

**Table 17 Comparing the results of scenario Pr12 in $E_5$, $E_7$, and $E_8$.**

| Experiment | Scenario code | Path |
|---|---|---|
| $E_5$ | Pr12 | 0 27 19 17 79 14 32 65 0 |
| $E_7$ | Pr12 | 0 6 52 4 35 87 0 |
| $E_8$ | Pr12 | 0 6 52 21 32 33 0 |
| *TOPTW* | Pr12 | 0 6 52 21 32 41 95 0 |

**Remark 6.** *The Happiness Model is designed to handle scalability through the integration of the Item Constraints Data Model (ICDM) and use of an advanced optimization algorithms such as Ant Colony Optimization (ACO). The ICDM model is employed to reduce data dimensionality and search space. This can significantly improve the scalability of the model, especially for large datasets. In addition, the ACO algorithm has been employed to handle complex itineraries efficiently. As dataset size increases, the ACO algorithm adapts by optimizing the search space through pheromone-based learning. The conducted experiments show that the model scales effectively when applied to larger datasets such as those in the TOPTW benchmark, with minimal loss in computational efficiency. For more complex itineraries, where travelers wish to visit multiple POIs across multiple days, the model maintains its efficiency by optimizing time and connection constraints simultaneously. Future work will explore the use of parallel computing and cloud-based infrastructures to further enhance scalability.*

**Remark 7.** *The Happiness Model (HM) faces several computational challenges, including the complexity of the optimization problem and the computational demands for large datasets or real-time applications. To address these challenges, an efficient ACO algorithm is implemented and explored scalability improvements using the ICDM model. This combination of strategies has made the HM more practical and applicable for real-world use cases.*

# CONCLUSION

A novel personalized happiness model has been proposed in this study to tailor recommended trips based on individual traveler preferences. It is worth mentioning that the proposed happiness model is the first dynamic model capable of successfully customizing tour itineraries based on specific components of the trip, namely: activities, connections, and waiting times. The experimental results indicate that the proposed model significantly enhances the personalization of tour itineraries, outperforming existing models in terms of both effectiveness and intuitiveness. Future works include extending the proposed model to accommodate group travel preferences and dynamics, as well as conducting extensive field experiments and user studies. Additionally, collecting real-world user data and exploring alternative datasets are planned to be conducted to further validate and improve the performance of the proposed model.
## APPENDIX

In this section, a detailed results for the recommended tour trips are provided. Tables A-1–A-8 illustrate the full path for each experiments from experiment #1 to experiment #8 on Dataset 1 and Dataset 2 as per the recommended trip.

**Table A-1 The results of experiment #1 on Dataset 1; a full path for each scenario.**

| Code | Path |
|---|---|
| Pr02 | 0>6>22>32>95>57>29>58>52>61>45>26>21>91>46>54>11>25>69>81>0 |
| Pr03 | 0>69>62>139>5>104>29>82>59>93>33>128>123>18>22>25>43>63>0 |
| Pr04 | 0>147>31>4>23>44>37>175>8>124>83>136>54>61>121>104>152>117>65>5>0 |
| Pr05 | 0>44>80>29>81>211>4>154>75>157>193>11>91>203>142>235>187>20>55>35>197>181>28>9>32>0 |
| Pr06 | 0>80>149>278>116>74>279>95>210>130>219>123>241>196>246>129>166>228>167>215>33>182>121>197>286>0 |
| Pr08 | 0>117>87>108>68>125>77>132>106>133>127>109>102>140>10>42>23>28>19>0 |
| Pr09 | 0>141>96>50>49>160>195>102>76>156>21>118>16>73>108>127>207>197>170>5>0 |
| Pr10 | 0>19>103>206>246>263>108>251>22>277>92>201>192>100>179>233>17>177>173>144>129>274>157>118>14>54>0 |

**Table A-2 The results of experiment #2 on Dataset 1; a full path for each scenario.**

| Code | Path |
|---|---|
| Pr02 | 0>6>22>32>95>57>29>58>52>61>45>26>21>91>46>54>11>25>69>81>0 |
| Pr03 | 0>69>62>139>5>104>29>82>59>93>33>128>123>18>22>25>43>63>0 |
| Pr04 | 0>147>31>4>23>44>37>175>8>124>83>136>54>61>121>104>152>117>65>5>0 |
| Pr05 | 0>44>80>29>81>211>4>154>75>157>193>11>91>203>142>235>187>20>55>35>197>181>28>9>32>0 |
| Pr06 | 0>80>149>278>116>74>279>95>210>130>219>123>241>196>246>129>166>228>167>215>33>182>121>197>286>0 |
| Pr08 | 0>117>87>108>68>125>77>132>106>133>127>109>102>140>10>42>23>28>19>0 |
| Pr09 | 0>141>96>50>49>160>195>102>76>156>21>118>16>73>108>127>207>197>170>5>175>0 |
| Pr10 | 0>19>103>206>246>263>108>251>22>277>92>201>192>100>179>233>17>177>173>144>129>274>157>118>14>54>0 |

**Table A-3 The results of experiment #3 on Dataset 1; a full path for each scenario.**

| Code | Path |
|---|---|
| Pr02 | 0>6>32>60>72>95>57>29>41>61>52>21>91>46>54>11>25>87>33>81>0 |
| Pr03 | 0>69>39>62>17>5>104>29>82>59>93>33>128>123>18>22>25>4>12>0 |
| Pr04 | 0>147>31>4>23>51>185>175>8>37>100>169>96>61>191>26>144>67>152>0 |
| Pr05 | 0>44>8>24>116>29>102>188>183>105>165>18>132>79>85>182>30>198>64>225>121>28>9>32>0 |
| Pr06 | 0>258>44>265>39>180>118>179>111>184>11>172>261>150>273>78>228>33>215>182>121>167>252>286>0 |
| Pr08 | 0>117>87>125>77>139>108>134>132>65>24>92>1 >76>112>124>21>138>41>144>0 |
| Pr09 | 0>171>180>59>105>42>138>113>204>179>65>108>16>121>127>73>207>5>170>51>116>181>74>175>0 |
| Pr10 | 0>19>188>143>166>92>32>25>114>97>267>232>205>34>151>134>141>49>86>126>51>54>14>83>0 |

**Table A-4  The results of experiment #4 on Dataset 1; a full path for each scenario.**

| Code | Path |
| --- | --- |
| Pr02 | 0>6>18>67>80>24>32>65>57>26>35>31>45>52>21>91>54>88>33>78>87>81>0 |
| Pr03 | 0>69>39>87>17>139>52>88>80>26>71>107>43>22>123>18>25>4>0 |
| Pr04 | 0>147>31>4>23>175>8>37>44>185>189>9>177>121>67>161>117>65>152>0 |
| Pr05 | 0>44>80>29>81>231>115>59>102>188>183>105>132>85>182>212>30>64>198>136>108>9>32>103>0 |
| Pr06 | 0>80>267>195>219>16>130>138>229>95>188>127>261>87>78>14>197>287>143>228>167>201>0 |
| Pr08 | 0>117>87>108>68>77>125>56>135>54>123>24>92>81>144>71>111>28>101>0 |
| Pr09 | 0>171>166>192>90>138>89>134>183>114>51>5>108>73>127>121>16>118>156>21>31>74>170>0 |
| Pr10 | 0>19>263>108>181>115>253>158>243>27>2>258>72>155>242>71>59>101>285>156>208>6>152>128>0 |

**Table A-5  The results of experiment #5 on Dataset 2; a full path for each scenario.**

| Code | Path |
| --- | --- |
| Pr11 | 0>29>34>21>3>8>25>41>16>10>1>27>35>2>28>40>44>38>9>12>0 |
| Pr12 | 0>27>19>17>79>14>32>65>0 |
| Pr13 | 0>66>40>29>64>78>9>122>44>117>18>105>123>23>99>31>115>10>63>129>79>104>141>0 |
| Pr14 | 0>114>96>60>35>179>174>26>88>76>178>180>9>168>120>139>84>91>95>138>184>25>85>147>183>0 |
| Pr15 | 0>169>103>69>165>80>107>4>11>150>119>44>237>2>145>203>71>48>136>153>85>77>125>89>202>67>25>209>204>21>36>0 |
| Pr16 | 0>124>149>230>128>121>30>33>215>228>106>139>81>167>182>122>143>197>287>34>159>116>78>137>261>216>69>0 |
| Pr17 | 0>51>50>29>54>15>30>12>56>6>5>58>65>37>21>45>9>10>69>0 |
| Pr18 | 0>118>75>70>9>133>7>89>63>41>138>11>36>88>116>115>8>14>93>48>50>82>0 |
| Pr19 | 0>50>214>53>123>30>137>88>74>213>108>113>75>106>9>72>195>206>118>166>200>87>135>14>105>5>0 |
| Pr20 | 0>58>246>205>27>101>114>105>277>220>144>173>20>8>208>196>279>2>256>159>124>89>85>138>179>228>56>12>241>181>0 |

**Table A-6  The results of experiment #6 on Dataset 2; a full path for each scenario.**

| Code | Path |
| --- | --- |
| Pr11 | 0>29>34>21>3>8>25>41>16>10>1>27>35>2>28>40>44>38>9>12>0 |
| Pr12 | 0>27>19>17>79>14>32>65>0 |
| Pr13 | 0>66>40>29>64>78>9>122>44>117>18>105>123>23>99>31>115>10>63>129>79>104>141>0 |
| Pr14 | 0>114>96>60>35>179>174>26>88>76>178>180>9>168>120>139>84>91>95>138>184>25>85>147>183>0 |
| Pr15 | 0>169>103>69>165>80>107>4>11>150>119>44>237>2>145>203>71>48>136>153>85>77>125>89>202>67>25>209>204>21>36>0 |
| Pr16 | 0>124>149>230>128>121>30>33>215>228>106>139>81>167>182>122>143>197>287>34>159>116>78>137>261>216>69>0 |
| Pr17 | 0>51>50>29>54>15>30>12>56>6>5>58>65>37>21>45>9>10>69>0 |
| Pr18 | 0>118>75>70>9>133>7>89>63>41>138>11>36>88>116>115>8>14>93>48>50>82>0 |
| Pr19 | 0>50>214>53>123>30>137>88>74>213>108>113>75>106>9>72>195>206>118>166>200>87>135>14>105>5>0 |
| Pr20 | 0>58>246>205>27>101>114>105>277>220>144>173>20>8>208>196>279>2>256>159>124>89>85>138>179>228>56>12>241>181>0 |

**Table A-7  The results of experiment #7 on Dataset 2; a full path for each scenario.**

| Code | Path |
| --- | --- |
| Pr11 | 0>20>8>21>3>42>45>24>23>46>26>10>1>13>7>38>22>28>0 |
| Pr12 | 0>6>52>4>35>87>0 |
| Pr13 | 0>82>118>53>29>64>33>49>105>125>44>117>18>4>2>95>81>15>48>16>70>74>0 |
| Pr14 | 0>181>60>35>179>174>26>3>134>124>20>85>108>103>157>155>91>95>180>33>66>176>16>162>0 |
| Pr15 | 0>216>53>117>230>135>218>11>80>123>78>192>48>120>150>119>24>49>125>40>193>176>2>25>209>204>154>174>179>0 |
| Pr16 | 0>124>149>230>128>30>121>12>33>215>228>139>167>81>106>182>143>197>287>34>116>78>86>3>0 |
| Pr17 | 0>43>51>50>29>23>54>22>32>30>36>35>37>15>60>61>45>9>10>69>0 |
| Pr18 | 0>124>128>113>71>106>34>75>70>87>30>18>100>24>48>83>2>46>89>77>62>107>82>0 |
| Pr19 | 0>50>214>53>123>27>63>192>166>80>83>74>213>108>13>148>203>201>186>49>175>8>9>72>149>14>23>0 |
| Pr20 | 0>125>249>189>242>156>142>5>285>225>243>163>84>176>205>97>129>63>275>276>263>251>26>120>132>168>221>138>143>207>0 |

**Table A-8  The results of experiment #8 on Dataset 2; a full path for each scenario.**

| Code | Path |
| --- | --- |
| Pr11 | 0>20>8>21>3>42>27>35>2>29>43>37>7>30>9>26>33>22>28>0 |
| Pr12 | 0>6>52>21>32>33>0 |
| Pr13 | 0>134>29>24>111>107>26>13>82>4>2>95>44>117>18>7>130>61>86>58>70>74>0 |
| Pr14 | 0>114>65>127>123>152>124>12>168>120>81>179>174>26>188>157>39>91>95>126>151>13>104>61>183>0 |
| Pr15 | 0>216>211>150>215>120>184>80>117>231>11>182>163>165>62>74>107>49>88>75>119>77>36>79>142>61>149>193>0 |
| Pr16 | 0>124>149>230>128>121>30>33>215>228>106>167>81>182>122>143>287>197>34>116>78>137>261>168>253>3>0 |
| Pr17 | 0>5>47>44>43>63>35>1>22>32>30>62>15>60>61>45>9>10>69>0 |
| Pr18 | 0>125>17>54>18>131>32>1>84>61>115>8>30>116>34>70>87>39>105>92>99>0 |
| Pr19 | 0>131>107>88>117>184>76>80>35>50>214>116>96>195>206>137>49>8>9>72>165>105>5>68>58>21>93>0 |
| Pr20 | 0>7>208>14>161>101>141>134>267>72>83>63>275>173>4>184>219>65>119>140>191>263>251>23>71>233>11>228>110>143>207>0 |

### Funding

This study is supported *via* funding from Prince Sattam bin Abdulaziz University project number (PSAU/2025/R/1446). The funders had no role in study design, data collection and analysis, decision to publish, or preparation of the manuscript.

### Grant Disclosures

The following grant information was disclosed by the authors:
Prince Sattam bin Abdulaziz University: PSAU/2025/R/1446.

### Competing Interests

The authors declare that they have no competing interests.

## Author Contributions

- Mohammed Alatiyyah conceived and designed the experiments, performed the experiments, analyzed the data, performed the computation work, prepared figures and/or tables, authored or reviewed drafts of the article, and approved the final draft.

## Data Availability

The raw measurements and source codes are available in the Supplemental Files.

The datasets we used are available at: https://www.mech.kuleuven.be/en/cib/op.

## Supplemental Information

Supplemental information for this article can be found online at http://dx.doi.org/10.7717/peerj-cs.2837#supplemental-information.

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
