# Peer review of "A novel user-centric happiness model for personalized tour recommendations"

_PeerJ Computer Science, doi:10.7717/peerj-cs.2837_

## Round 0.1 · original submission · Major Revisions

After the first round of review, "Major revisions" is recommended for this paper. Although reviewers found this paper is interesting. But, some technical issues need to be addressed before this paper can be reconsidered for this journal, in particular, to compare the performance of the proposed technique with other published works and enhance the explanation of the results presented in the figures.

Reviewer 1 ·

Basic reporting

In this paper a personalized model called happiness model (HM) has been proposed. The paper has been written well but it has the following issues:
1. The abstruct must concise. It needs to be rewritten.
2. The contribution of the paper needs to be mentioned clearly in the introduction section.
3. The outline of the paper is not mentioned mentioned in the introduction section.
4. The proposed technique needs to be written in algorithmic form.
5. Figure 2 doesnot depict proper connectivity and representation.
6. There was not much information provided about the future scope of work.

Experimental design

The proposed technique needs to be compared with the existing techniques.

Validity of the findings

The implementation is ok but it is very difficult to interpret figure 8.

Cite this review as

·

Basic reporting

The article introduces a novel model called the Happiness Model (HM) designed to optimize personalized tour recommendations by maximizing user satisfaction. The reporting is clear and well-structured, providing a comprehensive background on the current state of travel recommender systems. The literature review is thorough, outlining the limitations of existing models and positioning the proposed HM as an advancement in the field. The paper includes detailed explanations of the mathematical models, algorithms used, and the integration of the Item Constraints Data Model (ICDM), enhancing the reader's understanding. Figures and tables are well-labeled and complement the textual descriptions, providing visual support for the experimental results.

Experimental design

The experimental design employs publicly available datasets to test the proposed HM against existing models, such as the Time-Dependent Orienteering Problem with Time Windows (TOPTW). The experiments are designed to evaluate the performance of the HM in terms of personalization, efficiency, and user satisfaction. The use of the Ant Colony Optimization algorithm for comparative analysis strengthens the validity of the results. However, some limitations in the experimental setup, such as the lack of real-world data or user feedback to validate the findings, could be addressed in future research. The design is methodical, with clear explanations of the parameters and metrics used, ensuring reproducibility.

Validity of the findings

The findings demonstrate that the HM significantly enhances the personalization of tour recommendations compared to existing models. The results show that the HM can effectively integrate various factors, such as activity, connection, and waiting times, to provide a more tailored travel experience. The comparative analysis with existing models underscores the HM’s advantages in optimizing user satisfaction dynamically over time. While the experiments validate the proposed model's effectiveness, further validation using real-world applications or user studies would strengthen the claims of enhanced user satisfaction and practical utility

Additional comments

Strong Foundation with Minor Gaps: The article presents a novel and well-structured approach to personalized tour recommendations, and its contributions are clear. However, the lack of real-world validation, user-centric evaluation, and considerations for diverse user constraints highlight minor gaps that need to be addressed.
Addressable Shortcomings: The shortcomings identified, such as the need for real-world user feedback, a broader comparative analysis, and a discussion on scalability, are not fundamental flaws in the methodology. These can be addressed with additional explanations, clarifications, or supplementary experiments in the revised manuscript.
Potential for Practical Impact: The model shows significant potential for practical application, and enhancing the discussion on real-world adaptability and user studies would strengthen the paper's relevance and impact.
Suggested Revisions:
Incorporate User Feedback: Suggest adding a brief user evaluation or simulated feedback scenario to validate the claims on user satisfaction.
Scalability Discussion: Recommend including a discussion on how the model would perform with larger datasets or complex itineraries.
Integration of Real-Time Data: Encourage the authors to address the potential for integrating real-time data sources, even if just theoretically, to enhance the model's dynamic capabilities.
Expand Comparative Analysis: Ask the authors to expand the comparative analysis with other recent models to provide a broader context of the model's performance.
Clarify Implementation Complexity: Request additional details on the practical implementation of the model, possibly with suggestions for simplifying its application for real-world use.

---

## Round 0.2 · Major Revisions

Although the authors have addressed the comments from the first round of review, additional concerns have been raised by the reviewers. Therefore, a 'Major Revision' is recommended for this round of review."

The paper still has several critical issues that need addressing. Firstly, the abstract lacks clear data and comparison results as conclusions. Figure 1 is confusing, with unclear group definitions and ambiguous numerical values, making it difficult to interpret. The introduction fails to provide necessary background, identify problems, propose solutions, and highlight their advantages, resulting in a disorganized presentation. The related work section is also poorly structured; it should analyze relevant references first and then compare them with the proposed work, emphasizing its advantages.

Evaluation metrics such as Recall rate, Accuracy rate, and F1 score are essential for assessing the recommendation algorithm's performance, yet they are missing. The use of Ant Colony Optimization (ACO) is unclear; it should be properly categorized as part of the proposed model or a comparative algorithm. Additionally, the paper does not adequately explain the experimental background, methods, or dataset features, nor does it clarify the significance of experimental results, particularly in Tables 3 to 9. Finally, reliance on previously collected datasets without real-world context weakens the study's credibility; conducting experiments in real-world tourism scenarios is advisable for better validation.

Reviewer 1 ·

Basic reporting

The paper looks fine.

Experimental design

Experimental design is proper.

Validity of the findings

The findings look valid.

Additional comments

All my comments have been addressed well.

Cite this review as

Reviewer 3 ·

Basic reporting

1. In the abstract, the data results and comparison results must be listed as the conclusion.
2. Figure 1 is confusing at this place. The author fail to match the description of the Figure with the Figure itself. What is Group A? What is Group B? There is no illustration in the Figure. And what's the meaning of the figures 4,6,2,9,5,etc.? It's Abrupt and weird.
3. In the introduction part, the author fails to list the following tips:
(1) The backgound of the research. (2) The problems that must be solved. (3) The solutions. (4) The advantages of the solutions. And so on. It seems that the introduction is chucked together, the logic is mess.
4. The description of Related Work is to some extent, mess. The authors could firstly list and analyze all the relevant references with the proposed work. Then use a new paragraph, or a comparison table to analyze the comparison between the previous work with the proposed work. And most importantly, the advantages of the proposed work must be emphasized.
5. Some evaluation indexes like Recall rate, Accuracy rate, F1 score, etc, are the basic indexes to evaluate the recommendation algorithm. Without these indexes, how the function and performance could be testified?
6. The Ant Colony Optimization is weird to use in the experiment. Is the ACO a part of the proposed model? Or is it a comparison algorithm which is used to make comparison? The latter experiment seems has no relationship with the ACO. If it is a part of the proposed model, it should be arranged in Modeling and Method section, not in the experiment. If it is a comparison algorithm, the following experiment should make comparison with it.
7. The author fails to interpret:(1)The backround of the experiment. (2) The method of the experiment. (3) What are E1, E2,..., E8? (4) The feature of the original dataset.
8. All the interpretations on Tables e.g. Table 3, Table 4,...,Table 9. etc., are confusing. It is extremely difficult for readers to understand the meaning of the results. A good paper as well as its good experiment must be able to help readers easily understand the intention, rather than making them unable to comprehend it. In other words, since the author did such many experiments with quite a lot of data results, it is hard to find out where is the significance of the experiment, and how the experiment solve the problems in the introduction.
9. Without real-world circumstance and tourism scenario, it is hard to convince readers by only using some perviously-collected dataset. It is better to perform the experiment in a real-world environment.

Experimental design

The comments are all in "Basic reporting".

Validity of the findings

The comments are all in "Basic reporting".

Additional comments

The comments are all in "Basic reporting".

Cite this review as

Reviewer 4 ·

Basic reporting

1. The usage of the English language should be improved in the paper.
2. The structure and format of the paper should be consistent throughout.
3. Avoid using personalised keywords like we , from the paper.

Experimental design

1. The abstract section needs improvement.
2. The technical depth of the paper is limited and should be improved.
3. Mathematical modelling of the paper should be enhanced.
4. The caption of the figures and table should be concise.
.

Validity of the findings

1. The validity of the results are in questions. A through comparison with the existing work should be incorporated in the paper.
2. Results presentation should be improved in the paper. e.g. Figure 8 shows a slant line without any legends and detail explanation.
3. The results presented in the figure 9 should be rechecked and improved.
4. Reference section needs improvements

Additional comments

Overall, The level of this paper is below the journal standard. Major problem of this paper is lack of novelty of the proposed method. The verification and authenticity are also questionable.

Cite this review as

---

## Round 0.3 · accepted · Accept

The author has thoroughly addressed the reviewers' comments, and the manuscript now meets the journal's standards for publication. I therefore recommend accepting it in its current form.

Reviewer 3 ·

Basic reporting

The authors have revised the manuscript in line with the comments. I have no further concern.

Experimental design

The authors have revised the manuscript in line with the comments. I have no further concern.

Validity of the findings

The authors have revised the manuscript in line with the comments. I have no further concern.

Additional comments

The authors have revised the manuscript in line with the comments. I have no further concern.

Cite this review as